# Can Large Language Models Match the Conclusions of Systematic Reviews?

## Abstract

Systematic reviews (SR), in which experts summarize and analyze evidence across individual studies to provide insights on a specialized topic, are a cornerstone for evidence-based clinical decision-making, research, and policy. Given the exponential growth of scientific articles, there is growing interest in using large language models (LLMs) to automate this process. However, the ability of LLMs to critically assess evidence and reason across multiple documents to provide expert-quality observations remains poorly characterized. We therefore ask: **Can LLMs match the conclusions of systematic reviews written by clinical experts when given access to the same studies?** To explore this question, we present MedEvidence, a benchmark pairing findings from SRs with the studies they are based on. We benchmark 24 LLMs on our MedEvidence dataset, including reasoning, medical specialist, and models of varying sizes. We find that reasoning does not necessarily improve performance, larger models do not consistently yield greater gains, and knowledge-based fine-tuning tends to degrade accuracy on MedEvidence. Instead, most models exhibit similar behavior: performance tends to degrade as token length increases, their responses show overconfidence, and all models show a lack of scientific skepticism toward low-quality findings. These results suggest that more work is still required before LLMs can reliably match the observations from expert-conducted SRs, even though these systems are already deployed and being used by clinicians.

## 1 Introduction

As the number of published articles grows exponentially [1], manually synthesizing findings from multiple sources has become highly time-consuming. Thus, there is growing interest in developing automatic tools to process, synthesize, and extract insights from scientific literature [2, 3]. In particular, large language model (LLM)-based systems could offer a promising solution for supporting and automating tasks such as conducting systematic reviews (SRs). For example, several LLM-assisted tools such as Deep Research [4, 5], Elicit [6], and Open Evidence [7], have already been deployed. The momentum behind these technologies is further exemplified by the U.S. Food and Drug Administration's launch of an LLM-assisted scientific review pilot on May 2025 [8].

However, despite multiple deployments and efforts assessing scientific synthesis generation, the behavior of LLMs across key variables that influence generation remains poorly understood. In particular, their ability to synthesize findings from multiple studies—each varying in study type, population size, and risk of bias—and to navigate conflicting evidence (as medical findings can often contradict one another) is not well-characterized. Understanding these behaviors is essential, as medical knowledge is continually reshaped by new clinical trials, cohort studies, and expert opinions. Thus, like medical professionals do, LLMs must be capable of integrating the latest findings (e.g.

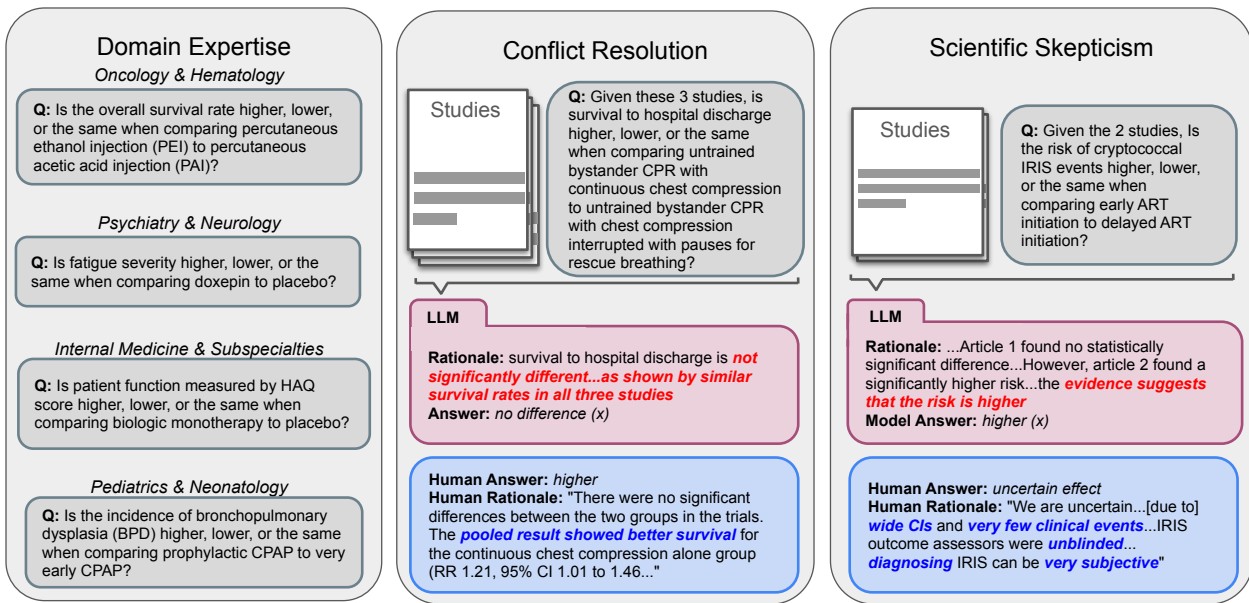

Figure 1: Core skills evaluated by MedEvidence including: medical domain expertise across 10 different specialties, synthesizing conflicting evidence, and applying scientific skepticism when studies exhibit a high risk of bias (e.g. due to small sample sizes or insufficient supporting evidence).

via retrieval augmentation) [9], weighing the strength of varying evidence, and applying appropriate skepticism when needed to produce reliable, up-to-date recommendations (as shown in Figure 1).

While prior work has successfully evaluated LLMs on their internal "static" medical knowledge [10, 11], assessing LLMs' capability to reason across multiple sources and draw expert-level conclusions remains a significant challenge. Specifically, previous efforts have often evaluated LLMs' ability to generate summaries on a given topic. This approach requires a thorough review of every detail in the generated content and lacks easily verifiable ground truth; therefore, medical experts are typically needed to assess output accuracy [12, 13, 14, 15, 16], making evaluation time-consuming and hard to scale. To address this, we remove the complexity of evaluating long-format summaries and retrieving relevant papers to pose an even simpler, but fundamental question: **Can LLMs replicate the individual conclusions of expert-written SRs when provided with the same source studies?** We explore this question in a controlled setting by collecting open-access SRs along with their associated reference articles. We then extract individual findings and reformat them into a closed question-answering (QA) task to enable straightforward evaluation. This allows us to test whether LLMs, when provided with the same evidence selected by experts, can reproduce each conclusion. To this end, we introduce the following contributions:

- **MedEvidence Benchmark** We introduce MedEvidence, a human-curated benchmark of 284 questions curated from the conclusions of 100 open-access SRs across 10 medical specialties. Each question evaluates comparative treatment effectiveness on clinical outcomes. All questions are manually transformed into closed-form question answering to enable large-scale evaluation. In addition, human annotators extract evidence quality (based on the SR's analysis), determine whether full-text access is necessary, and collect the relevant sources needed to replicate the SR findings.

- **Large-scale evaluation on MedEvidence** We leverage MedEvidence to perform an in-depth analysis of 24 LLMs spanning general-domain, medical-finetuned, and reasoning models. By utilizing MedEvidence's metadata, we dissect and examine success and failure modes, helping to identify targeted directions for future work.

Table 1: Comparison of factuality and evidence reasoning benchmarks with medical focus. We compare MedEvidence to prior datasets across attributes relevant to systematic review-style reasoning. MedEvidence is the only dataset to satisfy all criteria.

| Dataset | Size | Topic | Curation | Expert-Grounded Answer | Automated Evaluation | Multiple Sources | Evidence Quality | Source-Level Concordance |
|---------|------|-------|----------|:---:|:---:|:---:|:---:|:---:|
| Reason et al. | 4 | Medicine | Human | ✓ | ✗ | ✓ | ✗ | ✗ |
| Schopow et al. | 1 | Medicine | Human | ✓ | ✗ | ✓ | ✗ | ✗ |
| MedREQAL | 2786 | Medicine | LLM | ✓ | ✓ | ✗ | ✓ | ✗ |
| HealthFC | 750 | Consumer Health | Human | ✓ | ✓ | ✗ | ✓ | ✗ |
| ConflictingQA | 238 | Multi-Domain | LLM | ✗ | ✗ | ✓ | ✗ | ✓ |
| MedEvidence | 284 | Medicine | Human | ✓ | ✓ | ✓ | ✓ | ✓ |

## 2 Related work

An overview of related works and the key distinct contributions of our current work are summarized in Table 1.

**LLM-based medical systematic review** Numerous studies have explored the potential of LLMs to automate various aspects of scientific literature review, including literature search, query augmentation, screening, data extraction, bias assessment, narrative synthesis, and answering simple clinical inquiries [17, 18]. However, larger-scale evaluations of LLM-based SR or meta-analyses generation remain relatively underexplored. Reason et al. [12] examined the ability of LLMs to extract numerical data from abstracts and generate executable code to perform meta-analyses. While their results are promising, the study is limited to just four individual case studies. Schopow et al. [13] and Qureshi et al. [14] investigate LLM usage across a range of systematic review stages, including meta-review and narrative evidence synthesis, but also present findings on a very small-case study scale ($N < 10$) and rely on comparison to humans. Overall, these investigations have been limited in scope and require substantial amounts of review from medical experts, highlighting the need for automated benchmarks to help evaluate LLMs' progress.

**Verification of medical facts derived from systematic reviews** Several studies have leveraged SRs to benchmarked LLMs' ability to perform medical fact verification, where a model must decide whether to support or refute a given claim. For instance, MedREQAL [19] is an LLM-curated closed QA dataset designed to investigate how reliably models can verify claims derived from Cochrane SRs. However, it does not provide the sources used by the SRs. Instead, the dataset evaluates models on their internal knowledge, making the task a form of fact recall. HealthFC [20], on the other hand, tasks models with verifying claims analyzed by the medical fact-checking site Medizin Transparent, but it only provides pre-synthesized analysis from the web portal as evidence. In contrast to real SR generation, this task primarily involves retrieving information from a pre-synthesized source, removing the real complexity of reasoning across unsynthesized evidence. Unlike prior work, MedEvidence requires extracting, reasoning over, and synthesizing relevant information across single or multiple sources (each with different levels of evidence) to match the expert-derived conclusion of a SR (without access to the original SR itself). It resembles the intricacies of SR analysis, as the raw sources (articles/abstracts) are directly provided to the model.

**LLM Behavior in the Presence of Conflicting Sources** ConflictingQA [21] examines how models respond to conflicting arguments supporting or refuting a claim. However, it focuses on inherently contentious questions without definitive answers, spans domains beyond medicine, and uses diverse online sources rather than peer-reviewed literature. ClashEval [22] investigates conflicts between a model's internal knowledge and external evidence, including a drug-related (medical) subset, but limits evaluation to single-source conflicts with artificially perturbed values. ConflictBank [23] and KNOT [24] assess model performance on specific conflict types—such as temporal inconsistencies, misinformation, and logic-based contradictions—but rely on factoid-style questions sourced from Wikipedia.These benchmarks only leverage relatively small and synthesized inputs.

To the best of our knowledge, no existing studies or datasets provide richly annotated data to systematically benchmark models' ability to align with the conclusions of medical systematic reviews while using the same underlying research documents as the original medical experts.

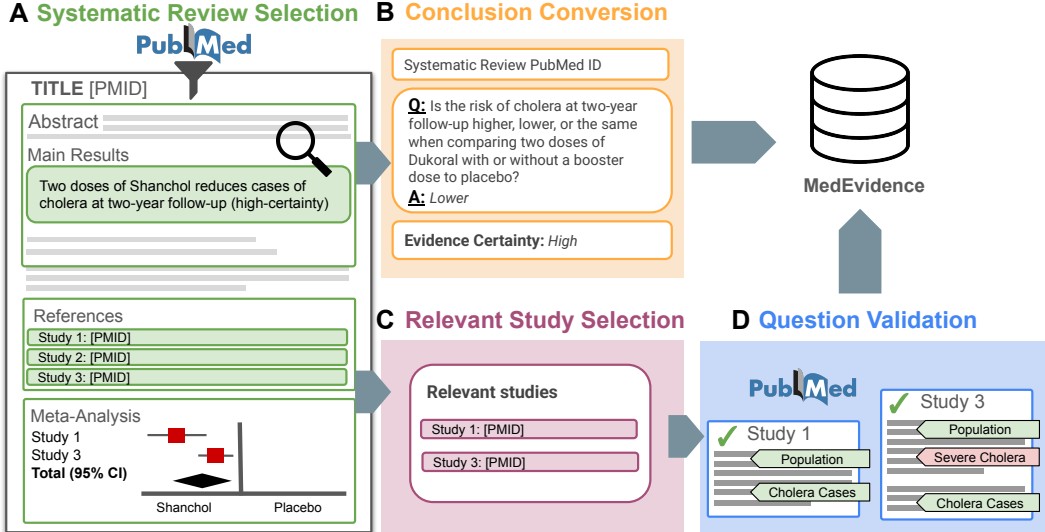

Figure 2: Overview of the dataset curation process for MedEvidence.

## 3 Dataset Curation Process

**Data provenance** We collect open-source systematic reviews, available via PubMed, conducted by Cochrane, an international non-profit organization dedicated to synthesizing evidence on healthcare interventions through contributions from over 30,000 volunteer clinician authors [25]. Cochrane is a long-standing and widely respected source of clinical evidence [26, 27], offering open-access content and analyses presented in a standardized format. Additionally, for each SR, we collect all the cited studies that are relevant for a given conclusion (we refer to these studies as 'sources'). When the source article's full text is available (i.e. the article is open-source), we obtain it using the existing BIOMEDICA dataset [28]; otherwise, abstracts are retrieved directly via PubMed's Entrez API [29]. All retrieved full-text articles use a CC-BY 4.0 license, which allows for re-distribution.

**Dataset curation pipeline** The core challenge in creating our dataset is ensuring that an LLM is provided with sufficient information to reproduce a given conclusion. To ensure a high-quality dataset, we developed a four-stage pipeline consisting of: (1) systematic review selection, (2) conclusion to questions conversion, (3) relevant study selection, and (4) question feasibility validation (as shown in Figure 2).

1. **Systematic review selection** We use Entrez to retrieve all Cochrane SRs published between January 1, 2014 to April 4, 2024 [30]. We only include systematic reviews for which all sourced studies are indexed in PubMed (with at least an abstract available). We additionally retrieve all data and metadata for the sourced studies, including: full-text via BIOMEDICA (when it is available), abstract, mesh terms, title, and publish date.

2. **Conclusion to question conversion.** Cochrane reviews follow a standardized format, allowing for a systematic conversion process. To identify potential questions, we followed the protocol below: Human annotators were instructed to review the SR abstract and examine the "Main Results" subsection (see Appendix Figure 9 for an example) to identify individual conclusive statements that statistically compare an intervention with a control group. These individual statements were then converted into question–answer pairs by the annotators, with answers belonging to a fixed set of classes. To be clear, `insufficient data` was used for statements by the SR authors explicitly indicating that no study investigated—or included sufficient data to analyze—the combination of treatment, control, and outcome; `uncertain effect` referred to cases where analysis was performed but definitive conclusions could not be made (see Appendix Section B.3 for more conversion details). Evidence certainty was extracted only when it was explicitly provided by the original SR authors, who use the standardized GRADE framework [31] to assess the quality of evidence in the included

studies. This certainty is often stated in the abstract, indicating the strength or quality of each observation.

3. **Relevant study selection** To identify relevant studies for a given SR, annotators used the analysis section provided in the appendix, which "weighs" the contributions of sources supporting each conclusion. For questions with insufficient data (where it is not possible to determine weights), reviewers were instructed to include studies cited in the SR that either (1) discuss the specified treatment and control but not the outcome, or (2) evaluate the treatment and outcome but compare against a different control.

4. **Question feasibility validation** Finally, given the question–answer pair and the source studies, annotators were tasked with determining whether the question was answerable based on the provided information. A question was considered answerable if at least 75% of the total weight in the analysis came from "valid" studies included in the meta-analysis. We define a study as "valid" if it (1) provides numerical data on both the intervention and control groups specified in the question, and (2) includes statistical or numerical details about the difference between the groups on the specified outcome—such as raw counts, p-values, confidence intervals, or risk ratios. The most common reason for discarding conclusions was when review authors pooled outcome data across studies, but the outcome was omitted or discussed without clear statistical detail in the abstracts of relevant studies.

In addition to these human-curated metadata, we use an LLMs to assess the percentage of individual source studies whose answer to the question aligns with the final answer provided in the systematic review. Thus, to calculate source-level agreement (which we call 'source concordance') we prompt DeepSeekV3 (the strongest model in our benchmark) to answer the question using only one single relevant source; the source is deemed to 'agree' with the final answer if and only if the LLM's classification with the one source matches the ground truth classification.

**Medical domain taxonomy assignment** To identify the relevant medical specialties in our dataset, we extract the Medical Subject Headings (MeSH terms)—a controlled vocabulary used by PubMed to index papers—from the 100 systematic reviews included in our dataset. We then feed this list into DeepSeek to generate a simplified categorization of specialties, resulting in 10 categories. Finally, we prompt DeepSeek to assign each question to the most relevant category, or to an "Other" category if no specific specialization is applicable.

## 4   Dataset Description

Table 2: Sample question from the dataset. Fields marked with an asterisk (*) use LLMs to assist the generation. Relevant source details are omitted here for brevity.

| Question | Is stroke prevention higher, lower, or the same when comparing Transcatheter Device Closure (TDC) to medical therapy? |
|---|---|
| **Answer** | no difference |
| **Relevant Sources (PubMed IDs)** | 22417252, 23514285, 23514286 |
| **Systematic Review (PubMed ID)** | 26346232 |
| **Review Publication Year** | 2015 |
| **Evidence Certainty** | n/a |
| **Open-Access Full-Text Needed** | no |
| ***Source Concordance** | 1.0 |
| ***Medical Specialty** | Surgery |

MedEvidence contains a total of 284 questions derived from 100 systematic reviews with 329 referenced individual articles, of which 114 have full-text available (see Appendix Figure 8 for a cohort diagram of the dataset). Questions were systematically collected by three human annotators with between one and five years of graduate education. Figure 3 shows the dataset distribution stratified by specialty, outcome effect, and source concordance with the expert-assessed treatment outcome effect (i.e. the correct answer). The benchmark covers topics from 10 medical specialties (e.g. public health, surgery, family medicine, etc.), five different outcome effects (`higher`, `lower`, `no difference`, `uncertain effect`, `insufficient data`), and three broad levels of concordance between the source paper and the correct answer (full agreement, no agreement, mixed agreement). Additional characteristic distributions of the dataset can be found in Appendix Figure 17.

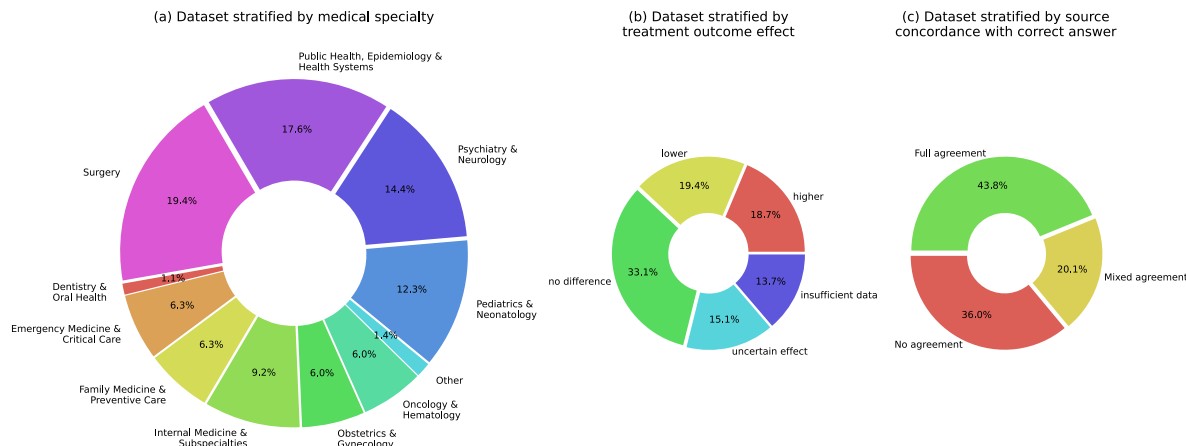

Figure 3: Key statistical characteristics of the questions in MedEvidence. (a) shows the dataset distribution stratified by medical specialty. (b) presents the distribution stratified by outcome effect. (c) shows the distribution stratified by source concordance with the expert-assessed treatment outcome effect (i.e. the correct answer).

**Data format.** MedEvidence is grouped by question; each question includes core data for evaluation, metadata, as well as the content details for the relevant sources. The core data consists of: a human-generated question of the form "Is [quantity of medical outcome] higher, lower, or the same when comparing [intervention] to [control]?"; the taxonomized answer to the question (`higher`, `lower`, `no difference`, `uncertain effect`, `insufficient data`); and the list of relevant studies (sources) used by the review authors to perform the analysis, identified by their unique PubMed IDs. We additionally provide the following metadata: the systematic review from which the question was extracted; the publication year of the systematic review; the authors' confidence in their analysis, also referred to as the 'evidence certainty' (`high`, `moderate`, `low`, `very low`, or `n/a` if not provided); a Boolean identification of whether full-text is available and needed to answer the question; the exact fractional source concordance; and the medical specialty associated with the question. Separately, for each source, we provide the unique PubMed ID, title, publication date if available, and content (full-text if available in PMC-OA, abstract otherwise). An individual data point example is shown in Table 2.

## 5 Benchmarking LLM performance

### 5.1 Experimental settings

**LLM selection** We selected 24 LLMs across different configurations, including a variety of sizes (from 7B to 671B), reasoning and non-reasoning capabilities, commercial and non-commercial licensing, and medical fine-tuning. This selection includes GPT-o1 [32], DeepSeek R1 [33], OpenThinker2 [34], GPT-4.1 [35], Qwen3 [36], Llama 4 [37], HuatuoGPT-o1 [38], OpenBioLLM [39], and more (please see Appendix Table 3 to see details of all selected models). This selection is non-exhaustive; rather, it is designed to investigate overarching trends across different model types.

**Prompting setup**

1. **Basic prompt** We evaluated all models in a zero-shot setting, prompting them to first provide a rationale for their answer, followed by an 'answer' field containing only one option from the list of five valid treatment outcome effects (higher, lower, no difference, uncertain effect, or insufficient data). To assess the models' "natural" behavior, we provided minimal guidance in the prompt beyond specifying the required response format, and supplied the abstracts or full text of the relevant studies as context (see Appendix Figure 11).

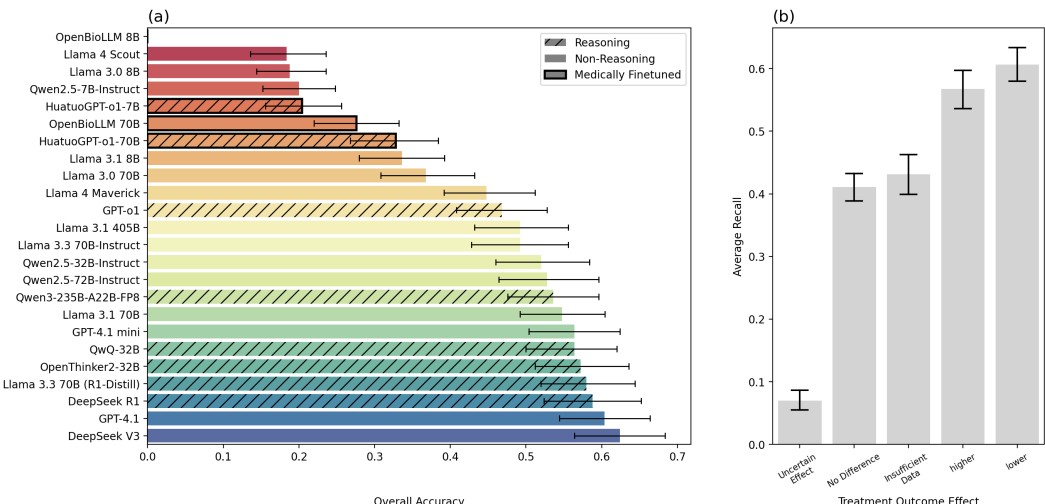

Figure 4: (a) Average model accuracy (and 95% interval) on MedEvidence. (b) Average recall by ground truth treatment outcome effect, aggregated across all models (with overall 95% interval). Per-model average recall by treatment outcome effect can be found in Appendix Figure 19.

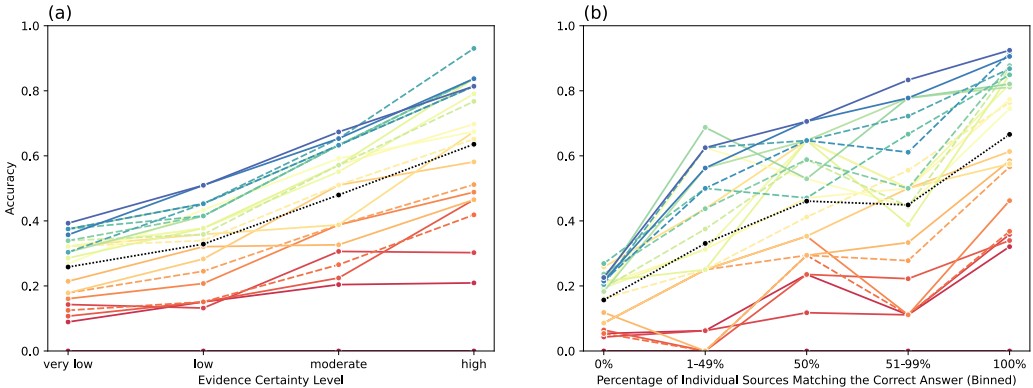

Figure 5: (a) Accuracy as a function of evidence certainty, shows a monotonically increasing trend. (b) Accuracy as a function of source concordance, defined as the percentage of relevant sources that agree with the final systematic review (SR) answer, also exhibits a monotonically increasing trend.

2. **Expert-guided prompt** LLMs may not natively understand how to handle multiple levels of evidence, which can lead to unfair evaluations. To address this, we explicitly design a prompt that instructs the LLM to summarize the study design and study population, and to assign a grade of evidence based on established definitions of grades of recommendation (see Appendix Figure 12 for the full prompt).

For both cases, if the input exceeded the LLM's context window, we used multi-step refinement (via LangChain's RefineDocumentsChain [40]) to iteratively refine the answer based on a sequence of article chunks. All models were evaluated with zero temperature to maximize reproducibility.

**LLM evaluation** Model performance was evaluated using accuracy based on an exact match between the answer field and the ground truth. Model outputs were lower-cased and stripped of whitespace before comparison. If no 'answer' field was provided, or if its content was not an exact rule-based match with the correct answer, the output was deemed incorrect. Confidence intervals (CIs) were calculated via bootstrap (95%, N=1000) [41].

**Compute Environment** Experiments were performed in a local on-prem university compute environment using 24 Intel Xeon 2.70GHz CPU cores, 8 Nvidia H200 GPUs, 16 Nvidia A6000 GPUs, and

223 40 TB of Storage. Large-scale models that could not be run locally in this environment were queried
224 in the cloud using public APIs available from together.ai or OpenAI.

## 6 Discussion

226 As shown in Figure 4 (a), even frontier models such
227 as DeepSeek V3 and GPT-4.1 demonstrate relatively
228 low average accuracy of 62.40% (56.35, 68.45) and
229 60.40% (54.30, 66.50), respectively—far from satu-
230 rating our benchmark. We identify four key factors
231 that influence model performance on our benchmark:
232 (1) token length, (2) dependency on treatment out-
233 comes, (3) inability to assess the quality of evidence,
234 and (4) lack of skepticism toward low-quality find-
235 ings. Additionally, we found that (5) medical fine-
236 tuning does not improve performance, and (6) model
237 size shows diminishing returns beyond 70 billion pa-
238 rameters. We explore each of these factors in more
239 detail below using the basic prompt setup.

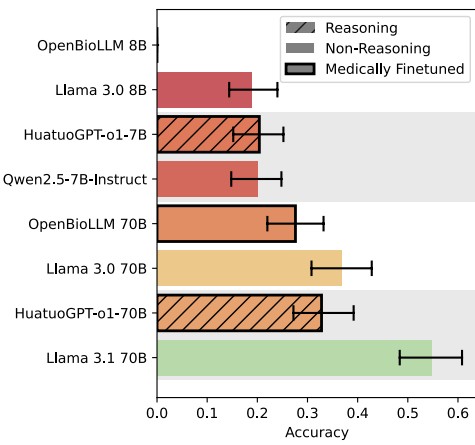

Figure 6: Medically-finetuned models vs their base generalist counterparts. Pairs of medical and base models are adjacent. 95% confidence intervals are calculated via bootstrapping with $N = 1000$.

240 **Reasoning vs non-reasoning LLMs** We highlight
241 that, in general, reasoning models do not consistently
242 outperform non-reasoning models of the same class
243 or size on MedEvidence (Figure 4 (a)), as evidenced
244 by DeepSeek V3 outperforming its reasoning counter-
245 part (DeepSeek R1), while LLaMA 3.3 70B distilled
246 from DeepSeek R1 outperforms the LLaMA 3.3 70B
247 base model.

248 **Model performance decreases as token length in-
249 creases** Generally, performance on MedEvidence drastically reduces as the number of tokens in-
250 creases (Appendix Figure 16). Naturally, training LLMs on long contexts does not guarantee improved
251 long-context understanding, as models may still struggle to utilize information from lengthy inputs
252 [42, 43].

253 **Model performance dependency on treatment outcome effect** Figure 4 (b) shows the per-class
254 recall stratified by treatment outcome effect. Overall, all models perform best on questions where the
255 correct answer corresponds to `higher` or `lower` effects—cases where a strong stance can be taken.
256 They are slightly less successful on `no difference` and `insufficient data` questions, where a
257 definitive conclusion is available but there is no clear preference for either treatment. Performance is
258 lowest on the most ambiguous class, `uncertain effect`. Notably, as shown in Appendix Figure 15,
259 models are generally reluctant to express uncertainty, often committing to a more certain outcome that
260 appears plausible. Notably, previous work has observed LLMs are verbally overconfident [44, 45]
261 and shown that reinforcement learning via human feedback (RLHF) amplifies this effect [46].

262 **Model performance improves with increasing levels of evidence** We leverage the evidence certainty
263 levels reported by experts in each systematic review (SR). As shown in Figure 5(a), the overall ability
264 of models to match SR conclusions improves as the level of evidence increases. We therefore explore
265 whether model performance is also associated with the level of source concordance. As shown in
266 Figure 5(b), models' ability to match human conclusions increases as the proportion of sources
267 agreeing with the correct answer increases (e.g., DeepSeek V3 achieves 92.45% accuracy at 100%
268 source agreement vs. 41.21% at 0% source agreement). This suggests that, unlike human experts,
269 current LLMs struggle to critically evaluate the quality of evidence and to remain skeptical of results.
270 We observe that this behavior persists even when models are prompted (using the expert-guided
271 prompt) to consider study design, population, and level of evidence (Appendix Figure 20).

272 **Medical finetuning does not improve performance** Figure 6 compares the average performance
273 of medically finetuned models to their base model counterparts. Across all comparisons, medical
274 finetuning fails to improve performance (even for medical-reasoning models) and, in most cases,
275 actually degrades it. Indeed, fine-tuning without proper calibration can harm generalization, some-

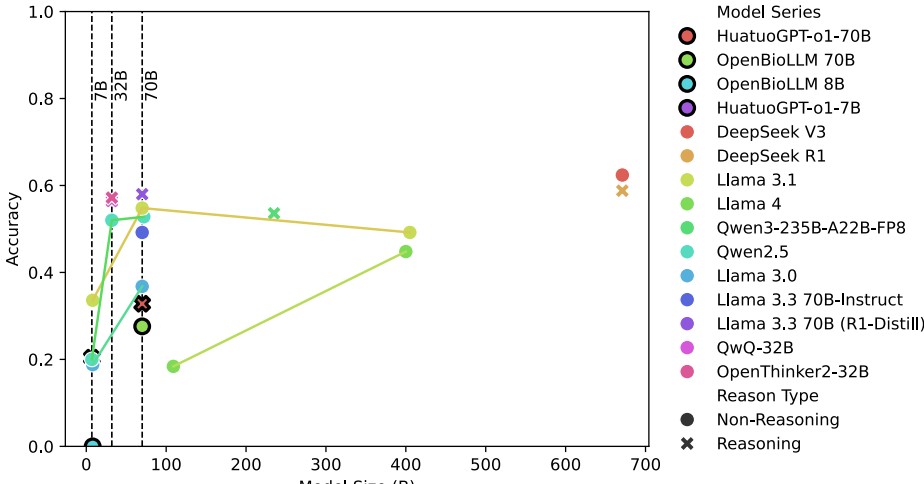

Figure 7: Average model accuracy on MedEvidence as a function of model size. We observe diminishing returns beyond 70 billion parameters.

times resulting in worse performance than the base model [47, 48, 49]. Similar behavior has been previously reported in long-context medical applications [11].

**Model size shows diminishing returns beyond 70B parameters** As shown in Figure 7, within the same model families, increasing size from 7B to 70B parameters yields substantial accuracy gains on MedEvidence. However, beyond this point, we observe rapidly diminishing returns, both within specific model families and across our suite of evaluated models more broadly.

Combined, our results suggest that synthesizing information across sources to match individual systematic reviews' conclusions eludes current scaling paradigms. Increasing test-time compute (i.e., reasoning) does not necessarily improve performance, larger models do not consistently yield greater gains, and knowledge-based fine-tuning tends to degrade performance. Instead, most models exhibit similar behavior: model performance tends to degrade as token length increases, their responses show overconfidence, and all models exhibit a lack of scientific skepticism toward low-quality findings. These results suggest that more work is still required before LLMs can reliably match the observations from expert-conducted SRs, even though LLM systems are already deployed and being used by clinicians.

**Limitations** Our study has several limitations. First, the dataset is subject to selection bias, as we only include a SR if all its sources are available (either full text/abstract). Second, while our benchmark is designed to isolate and provide a controlled environment to test LLMs' ability to reason over the same studies experts used to derive conclusions, it does not assess the full SR pipeline, including literature search, screening, or risk-of-bias assessment. Future work could incorporate multi-expert consensus or update findings based on newer studies to strengthen benchmark reliability.

## 7   Conclusion

Benchmarks drive advancements by providing a standard to measure progress and enabling researchers to identify weaknesses in current approaches. While LLMs are already deployed for scientific synthesis, our understanding of their failure modes still requires broader investigation. In this work, we present MedEvidence, a benchmark derived from gold-standard medical systematic reviews. We use MedEvidence to characterize the performance of 24 LLMs and find that, unlike humans, LLMs struggle with uncertain evidence and cannot exhibit skepticism when studies present design flaws. Consequently, given the same studies, frontier LLMs fail to match the conclusions of systematic reviews in at least 37% of evaluated cases. We release MedEvidence to enable researchers to track progress.

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
