# Appendix contents

# A  Societal impact

The use of large language models to automate systematic reviews offers clear potential to accelerate evidence synthesis in medicine and policy. However, when these systems produce incorrect or misleading results, clinicians and policymakers may base decisions on flawed findings, leading to inappropriate treatments or misguided recommendations.

Our study underscores the urgent need for continued research and cautious deployment. LLM-based systematic review systems need further rigorous validation, transparent uncertainty quantification, and mechanisms to detect and mitigate biases and errors. Only through careful development and oversight can these technologies be harnessed to benefit society without exacerbating existing risks or creating new harms.

# B  Dataset collection details

Below, we provide additional in-depth details regarding stages in dataset curation process.

## B.1  Systematic review selection

MedEvidence is originally derived from 6,709 Cochrane publications extracted via Entrez from PubMed. We first discarded any papers where first References subsection was not both entitled "Studies included in this review" and non-empty, as our initial extraction filter included Cochrane SR protocols and SRs finding no valid studies, which were not of interest. We filter for SRs where all included references have a retrievable abstract and limit to SRs with 12 or less references to reduce annotator burden and improve odds of finding SRs where questions can be validated. On average, the end-to-end creation of a single question requires approximately 20 minutes. Appendix Figure 8 presents a cohort diagram for the materialization of the dataset.

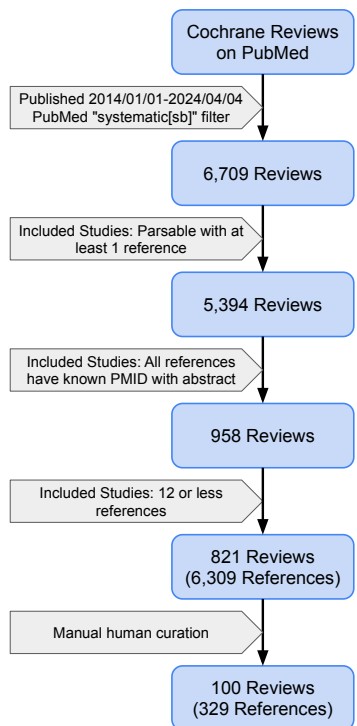

Figure 8: MedEvidence cohort diagram describing selection criteria for Cochrane SRs suitable for use in the MedEvidence dataset. Note that not all available papers in the second-to-last stage were manually reviewed for use in the final stage.

## B.2  Conclusion to question conversion

Appendix Figure 9 provides a direct example of a SR abstract parsed for manual question creation. We highlight the explicit statements ('conclusions') asserting differences between a treatment and control on an outcome, and the presence of standardized, author-provided assessment of evidence certainty for these individual conclusions. SR abstracts were consistently written in this form, allowing annotators

(VE), we converted the overall risk ratio (RR) to vaccine effectiveness using the formula: VE = (1 - RR) x 100%.

## Main results

Five RCTs, reported in 12 records, with 462,754 participants, met the inclusion criteria.

We identified trials on whole-cell plus recombinant vaccine (WC-rBS vaccine (Dukoral)) from Peru and trials on bivalent whole-cell vaccine (BivWC (Shanchol)) vaccine from India and Bangladesh. We did not identify any trials on other BivWC vaccines (Euvichol/Euvichol-Plus), or Hillchol.

Two doses of Dukoral with or without a booster dose reduces cases of cholera at two-year follow-up in a general population of children and adults, and at five-month follow-up in an adult male population (overall VE 76%; RR 0.24, 95% confidence interval (CI) 0.08 to 0.65; 2 trials, 16,423 participants; high-certainty evidence).

Two doses of Shanchol reduces cases of cholera at one-year follow-up (overall VE 37%; RR 0.63, 95% CI 0.47 to 0.85; 2 trials, 241,631 participants; high-certainty evidence), at two-year follow-up (overall VE 64%; RR 0.36, 95% CI 0.16 to 0.81; 2 trials, 168,540 participants; moderate-certainty evidence), and at five-year follow-up (overall VE 80%; RR 0.20, 95% CI 0.15 to 0.26; 1 trial, 54,519 participants; high-certainty evidence).

A single dose of Shanchol reduces cases of cholera at six-month follow-up (overall VE 40%; RR 0.60, 95% CI 0.47 to 0.77; 1 trial, 204,700 participants; high-certainty evidence), and at two-year follow-up (overall VE 39%; RR 0.61, 95% CI 0.53 to 0.70; 1 trial, 204,700 participants; high-certainty evidence).

A single dose of Shanchol also reduces cases of severe dehydrating cholera at six-month follow-up (overall VE 63%; RR 0.37, 95% CI 0.28 to 0.50; 1 trial, 204,700 participants; high-certainty evidence), and at two-year follow-up (overall VE 50%; RR 0.50, 95% CI 0.42 to 0.60; 1 trial, 204,700 participants; high-certainty evidence).

We found no differences in the reporting of adverse events due to vaccination between the vaccine and control/placebo groups.

## Authors' conclusions

Two doses of Dukoral reduces cases of cholera at two-year follow-up.

Two doses of Shanchol reduces cases of cholera at five-year follow-up, and a single dose of Shanchol reduces cases of cholera at two-year follow-up.

Overall, the vaccines were safe and well tolerated.

We found no trials on other BivWC vaccines (Euvichol/Euvichol-Plus). However, BivWC products (Shanchol, Euvichol/Euvichol-Plus) are considered to produce comparable vibriocidal responses. Therefore, it is reasonable to apply the results from Shanchol trials to the other BivWC products (Euvichol/Euvichol-Plus).

Figure 9: An example "Main Results" section from a Cochrane review used in MedEvidence (DOI: https://doi.org/10.1002/14651858.CD014573). Annotators were instructed to extract conclusions from this standardized sub-section of the SR abstract.

to consistently interpret the conclusion into a question. To define the correct answer to the generated question, annotators obeyed the following criteria:

- Outcomes, or pairs of treatments and controls, where the authors stated that no studies provided sufficient (or any) evidence to perform analysis were labeled as `insufficient data` questions.

- Conclusions in which the authors stated that there was "no difference" or "no significant difference" between treatments and controls were labeled as `no difference` questions.

- Conclusions where the authors stated a difference between outcomes either definitively or with qualification (e.g. 'X increases Y' or 'X may reduce Y') were given the appropriate `higher` or `lower` label.

- Conclusions where the authors expressed that uncertainty was too great to evaluate a treatment outcome were placed in the `uncertain effect` label class. Conclusions where authors assessed a difference, but then stated that they were very uncertain of their findings were deemed ambiguous and discarded.

**PICOs**

| Population (8) | Intervention (1) | Comparison (2) | Outcome (1) |
|---|---|---|---|
| Child, Preschool 2-5 years | Cholera Vaccines | Placebo | Cholera |
| Aged 80 and over 80+ | | Vaccinated | |

**B.3   Relevant study selection and question validation**

For author conclusions where more than one study was used, SRs provide meta-analyses over all relevant sources (an example meta-analysis is shown in Appendix Figure 10), allowing us to confirm whether the studies used in the original SR contain sufficient information to replicate the conclusions of human analysis.

| Study or Subgroup | log[Risk Ratio] | SE | BivWC (Shanchol) Total | Placebo Total | Weight | Risk Ratio IV, Random, 95% CI | Risk Ratio IV, Random, 95% CI |
|---|---|---|---|---|---|---|---|
| Bhattacharya 2013 | -1.43 | 0.11 | 30532 | 33466 | 50.8% | 0.24 [0.19 , 0.30] | |
| Qadri 2015 | -0.6 | 0.15 | 53170 | 51372 | 49.2% | 0.55 [0.41 , 0.74] | |
| **Total (95% CI)** | | | 83702 | 84838 | 100.0% | 0.36 [0.16 , 0.81] | |

Heterogeneity: Tau² = 0.33; Chi² = 19.91, df = 1 (P < 0.00001); I² = 95%
Test for overall effect: Z = 2.46 (P = 0.01)
Test for subgroup differences: Not applicable

Favours BivWC (Shanchol)    Favours placebo

Figure 10: An example meta-analysis from a Cochrane review (figure from DOI: https://doi.org/10.1002/14651858.CD014573). Notably, the set of relevant studies and their individual weighted contributions to the overall result are available.

# C  Additional dataset distributions

We present additional statistical characteristics of the questions in our MedEvidence dataset in Appendix Figure 11. We highlight that the dataset is balanced with respect to evidence certainty levels, strengthening the reliability of our main observations on the relationship between evidence certainty and model performance. With regard to the joint distribution of correct treatment outcome effect and evidence certainty, we note that the highly concentrated distributions for the `insufficient data` and `uncertain effect` classes are inherent to the nature of SR. For example, in the case of the `insufficient data` class, authors cannot draw definitive conclusions from analyses they were unable to perform; thus, their findings are most uncertain when the quality of evidence is poor.

# D  Evaluated models and prompts

The full list of 24 models we evaluate on MedEvidence is provided in Appendix Table 3. The exact prompt used to elicit LLM responses for evaluation under the basic prompt regime is provided in Appendix Figure 12. Under the expert-guided prompt regime, models were first instructed to generate a formatted article summmary using the summarization step (using Appendix Figure 13a), then asked to provide answers based on the generated summaries for all relevant articles (via Appendix Figure 13b). In all cases, chunks of original article text or previously-generated summarization were provided with a header line containing the article's title, date of publication (if available), and PubMed ID, allowing the LLM to recognize and assign blocks of content to different sources and synthesize in-context.

Table 3: List of evaluated models with their model size and context length limit we set for our experiments. Precision is 16-bit floating point unless specified otherwise.

| Model | Model Type | Parameter Sizes | Context Limit |
|---|---|---|---|
| DeepSeek R1 [33] | Generalist Reasoning | 671B | 131K |
| DeepSeek V3 [50] | Generalist Non-Reasoning | 671B | 131K |
| GPT-4.1 [35] | Generalist Non-Reasoning | Unknown | 1M |
| GPT-4.1 mini [35] | Generalist Non-Reasoning | Unknown | 131K |
| GPT-o1 [32] | Generalist Non-Reasoning | Unknown | 150K |
| HuatuoGPT-o1 [38] | Medical Reasoning | 7B, 70B | 32K, 16K |
| Llama 3.0 [51] | Generalist Non-Reasoning | 8B, 70B | 8K |
| Llama 3.1 [51] | Generalist Non-Reasoning | 8B, 70B, 405B | 131K |
| Llama 3.3 [51] | Generalist Non-Reasoning | 70B | 131K |
| Llama 3.3 (R1-Distill) [33] | Generalist Reasoning | 70B | 131K |
| Llama 4 Maverick [37] | Generalist Non-Reasoning | 400B (17B active) | 500K |
| Llama 4 Scout [37] | Generalist Non-Reasoning | 109B (17B active) | 1M |
| OpenBioLLM [39] | Medical Non-Reasoning | 8B, 70B | 8K |
| OpenThinker2 [34] | Generalist Reasoning | 32B | 131K |
| Qwen2.5 [52] | Generalist Non-Reasoning | 7B, 32B, 72B | 32K |
| Qwen3 [36] | Generalist Reasoning (hybrid) | 235B (22B active, 8-bit) | 32 K |
| QwQ [53] | Generalist Reasoning | 32B | 131K |

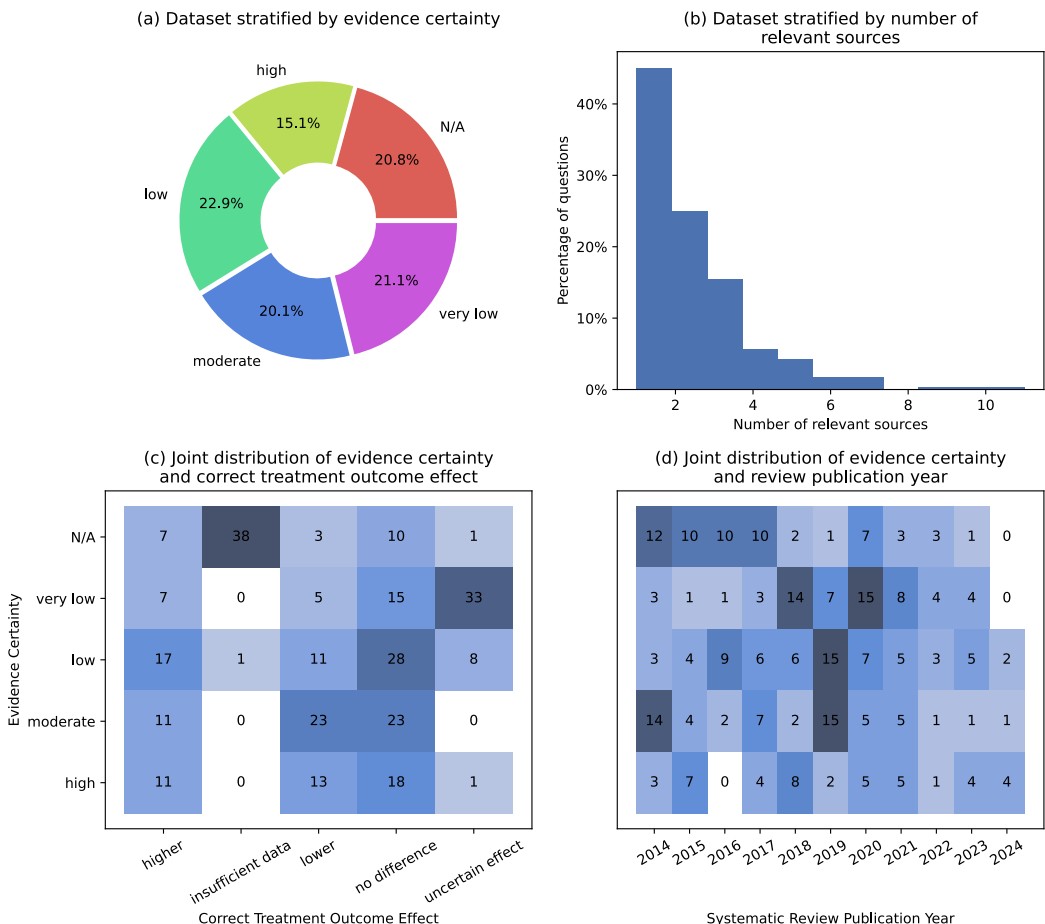

Figure 11: Additional statistical characteristics of MedEvidence. (a) shows the dataset distribution stratified by evidence certainty. (b) stratifies the questions by number of relevant sources. (c) is a joint distribution of evidence certainty and correct answer label. (d) shows the distribution of evidence certainties by systematic review publication year.

```
Given the ARTICLE SUMMARIES. Provide a concise and precise answer to the provided
    QUESTION.

After you think, return your answer with the following format:
- **Rationale**: Your rationale
- **Full Answer**: A precise answer, citing each fact with the Article ID in
    brackets (e.g. [2]).
- **Answer**: A final classification exactly matching one of the following options:
    Higher, Lower, No Difference, Insufficient Data, Uncertain Effect

Think step by step.
**QUESTION**: {question}
**ARTICLE SUMMARIES**: {context}
```

Figure 12: Prompt used to generate LLM responses to questions under the basic prompt setup.

```
You are the author of a Cochrane Collaboration systematic review, leveraging
    statistical analysis and assessing risks of bias in order to rigorously assess
     the effectiveness of medical interventions. As part of your review process,
    perform the following task:
As a subject expert, (1) summarize the evidence provided by a given ARTICLE as it
    pertains to a given QUESTION and (2) provide a possible answer.

Otherwise, if the provided article contains relevant information, you must return
    a list including the following items:

- **Study Design**: Type of study, level of evidence, and grade of recommendation
    according to the levels of evidence REC TABLE (provided Below).
- **Study Population**: Study size and patient population.
- **Summary**: A concise but comprehensive summary based on the previously
    specified information, with a focus on the main findings.
- **Possible Answer**: A concise feasible answer given the evidence.

**REC TABLE **: Levels of Evidence (from strongest [1a] to lowest [5]).

| Grade of Recommendation | Level of Evidence | Type of Study |
|--------------|--------------|------------------------------------------|
| A | 1a | Systematic review and meta-analysis of (homogeneous) randomized
    controlled trials |
| A | 1b | Individual randomized controlled trials (with narrow confidence
    intervals) |
| B | 2a | Systematic review of (homogeneous) cohort studies of 'exposed' and '
    unexposed' subjects |
| B | 2b | Individual cohort study / low-quality randomized control studies |
| B | 3a | Systematic review of (homogeneous) case-control studies |
| B | 3b | Individual case-control studies |
| C | 4 | Case series, low-quality cohort or case-control studies, or case reports
     |
| D | 5 | Expert opinions based on non-systematic reviews of results or
    mechanistic studies |"

Think step by step.
**QUESTION**: {question}
**ARTICLE TITLE**: {title}
**ARTICLE CONTENT**:
{context}
```

(a) Prompt used for the summarization step.

```
You are the author of a Cochrane Collaboration systematic review, leveraging
    statistical analysis and assessing risks of bias in order to rigorously assess
     the effectiveness of medical interventions. As part of your review process,
    perform the following task:
Given the ARTICLE SUMMARIES. Provide a concise and precise answer to the provided
    QUESTION.

After you think, return your answer with the following format:
- **Rationale**: Your rationale
- **Full Answer**: A precise answer, citing each fact with the Article ID in
    brackets (e.g. [2]).
- **Answer**: A final classification exactly matching one of the following options:
     Higher, Lower, No Difference, Insufficient Data, Uncertain Effect

Think step by step.
**QUESTION**: {question}
**ARTICLE SUMMARIES**: {context}
```

(b) Prompt used for the final answer step.

Figure 13: Prompts used to generate LLM responses to questions under the expert-guided prompt
setup, designed to attempt to explicitly enforce model awareness of evidence quality and strength.

## E    LLM instruction-following rates

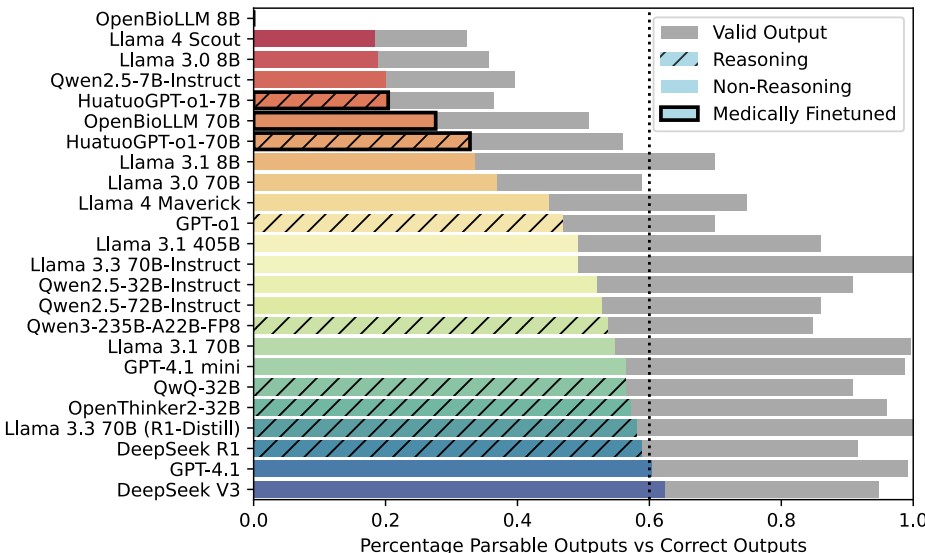

Figure 14: Per-model instruction-following rate, i.e. the percentage of questions for which the models produced a valid answer. Answers were valid only if the full "Answer" field in the model's final output exactly matched one of the defined answer classes (case-insensitive).

The rate at which LLMs provided valid answer output of any kind is shown in Appendix Figure 14. We note that a substantial portion of models exhibit a high rate of instruction-following failures: OpenBioLLM 8B and 70B; HuatuoGPT-o1 7B and 70B; Llama 4 Maverick and Scout; Llama 3.0 8B; and Llama 3.1 8B all fail to achieve a 60% instruction-following rate, and only Llama 3.3 70B (Instruct and R1-Distill) achieves perfect instruction-following. We highlight that OpenBioLLM 8B has a 0% instruction-following rate. Lastly, we observe that even when significant portion of the outputs are valid, models still have high error rates, with only an average of $58.1(\pm5.0)\%$ of valid model outputs being correct. These results demonstrate that, while a high instruction-following rate may diminish performance in small models, poor performance cannot be attributed to instruction-following errors alone.

## F    LLM performance as a function of number of relevant sources

As shown in Appendix Figure 15, we find no clear general trend between the number of relevant sources and model performance. Notably, this includes performance with a single relevant source (no model achieves even 60% accuracy), highlighting challenges in LLMs' ability to perform systematic review beyond resolving evidence conflicts. The only exceptions to this observation are the models with the overall poorest performance (colored in red and orange hues, such as HuatuoGPT-o1 7B and Llama 3.0 8B).

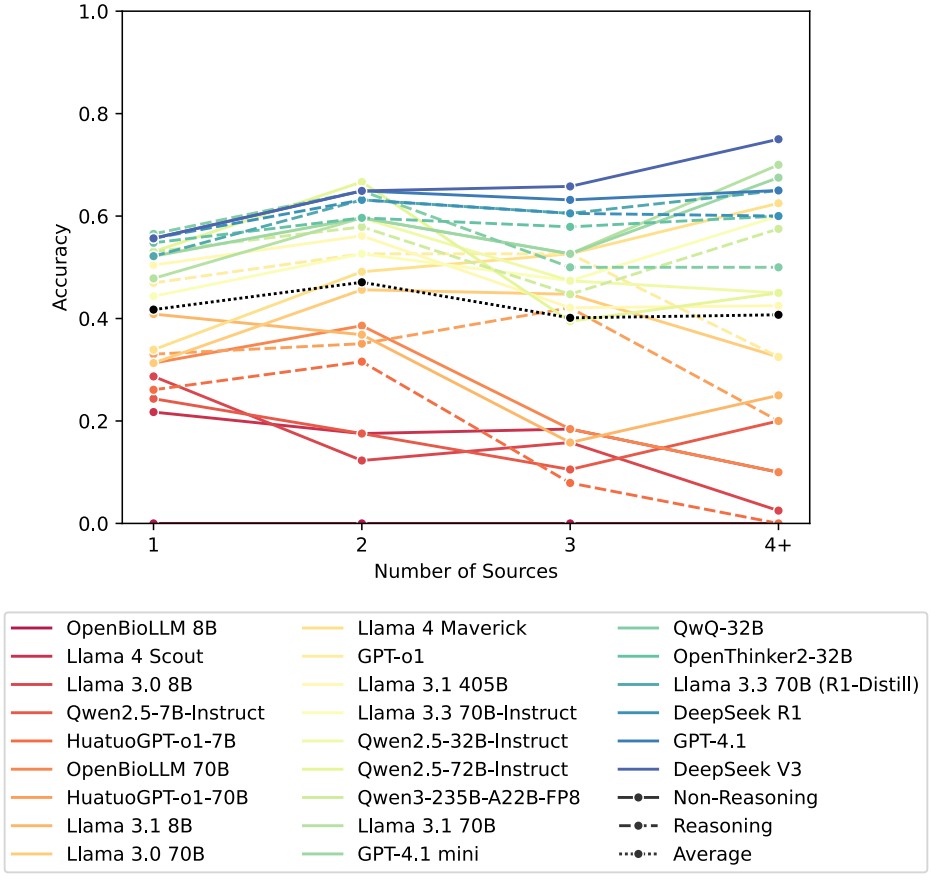

Figure 15: Model accuracy as a function of number of relevant sources.

# G    LLM performance as a function of token length of relevant sources

Given the lack of dependency on the number of sources on average accuracy, we directly investigate the dependency of model performance on the combined token length of all relevant sources; we present these results in Appendix Figure 16. As noted in the main analysis, performance consistently declines at high token counts, except for models with over 100B parameters. Notably, 32B models maintain over 50% average accuracy up to the 80–100% quantile (15K tokens and above). By contrast, 70–72B models fall below 50% accuracy around the 60–80% quantile (11–15K tokens). This decline in the 70–72B range is primarily driven by the underperformance of medically finetuned models (HuatuoGPT-o1 and OpenBioLLM).

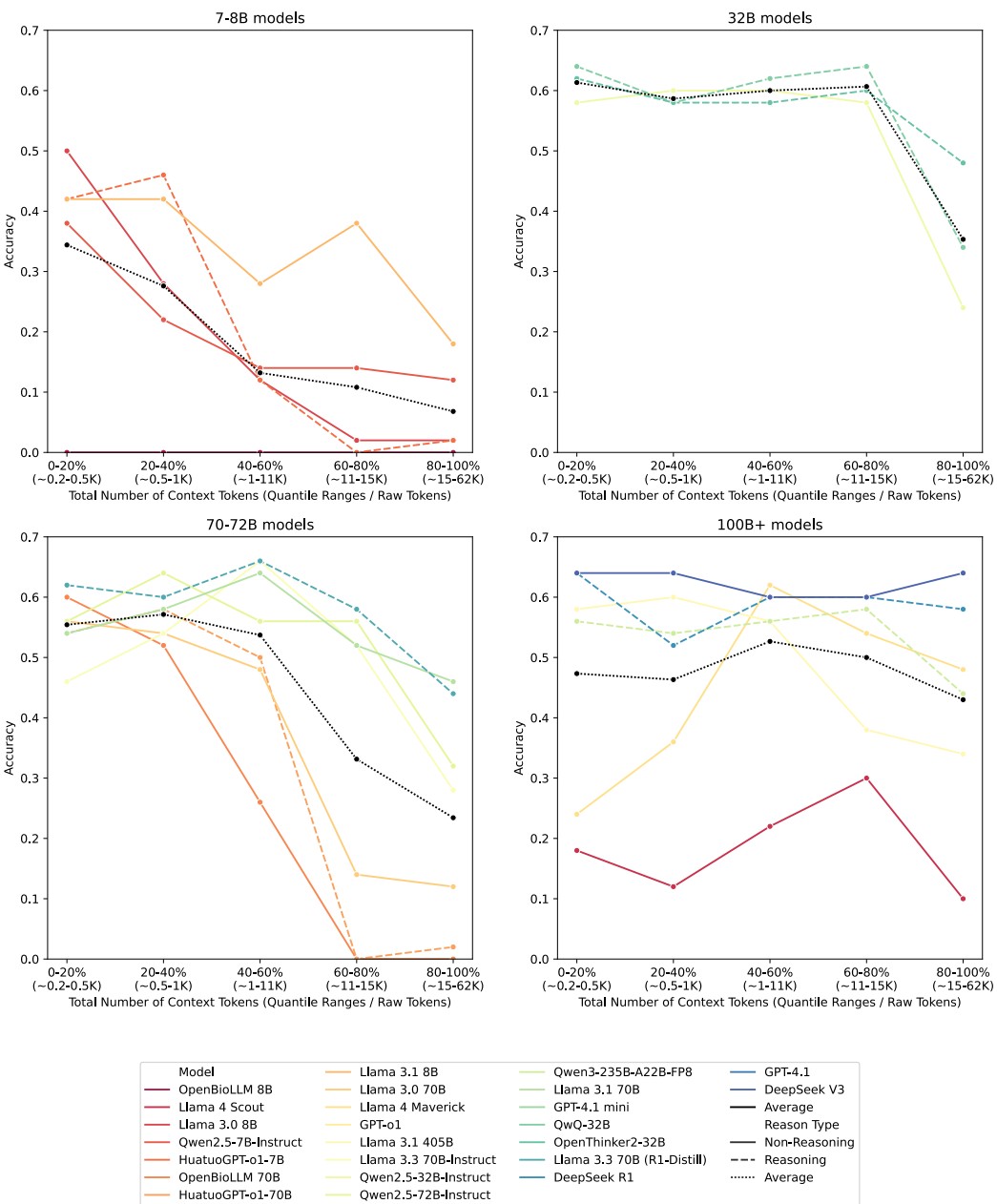

Figure 16: Model performance as a function of the number of tokens in the relevant studies, separated by model size range. Horizontal axis measures the accuracy by 5-quantiles.

## H  Average confusion matrices for treatment outcome effects

We assess which treatment outcome effect classes are most frequently misclassify by visualizing the confusion matrix averaged across all models. As shown in Figure 17, we observe that models with lower than 40% accuracy significantly skew the confusion matrix toward invalid outputs. However, when considering exclusively models with above 40% performance, we observe two significant trends. First, models are consistently unwilling to predict `uncertain effect`. Second, models consistently confuse the `uncertain effect` and `no difference` classes.

For completeness, we provide all individual confusion matrices in Appendix Section P.

## I  Performance by review publication year

As shown in Appendix Figure 18, performance steadily declines for more recent publication years, except for 2023 and 2024. These improvements may partially be explained by the fact that the majority of questions from 2024 involve high- or moderate-certainty evidence (as shown in Appendix Figure 11(d)); as a result, these questions are likely easier for models to answer.

## J  Per-class recall for individual models

We present individual model per-class recall in Appendix Figure 19. Notably, all models, without exception, perform poorly on the `uncertain effect` class. We highlight that Llama 3.3 70B-Instruct outperforms all other models on the `higher` and `lower` classes, but its overall accuracy is held back significantly by its poor performance on the `no difference` and `insufficient data` classes.

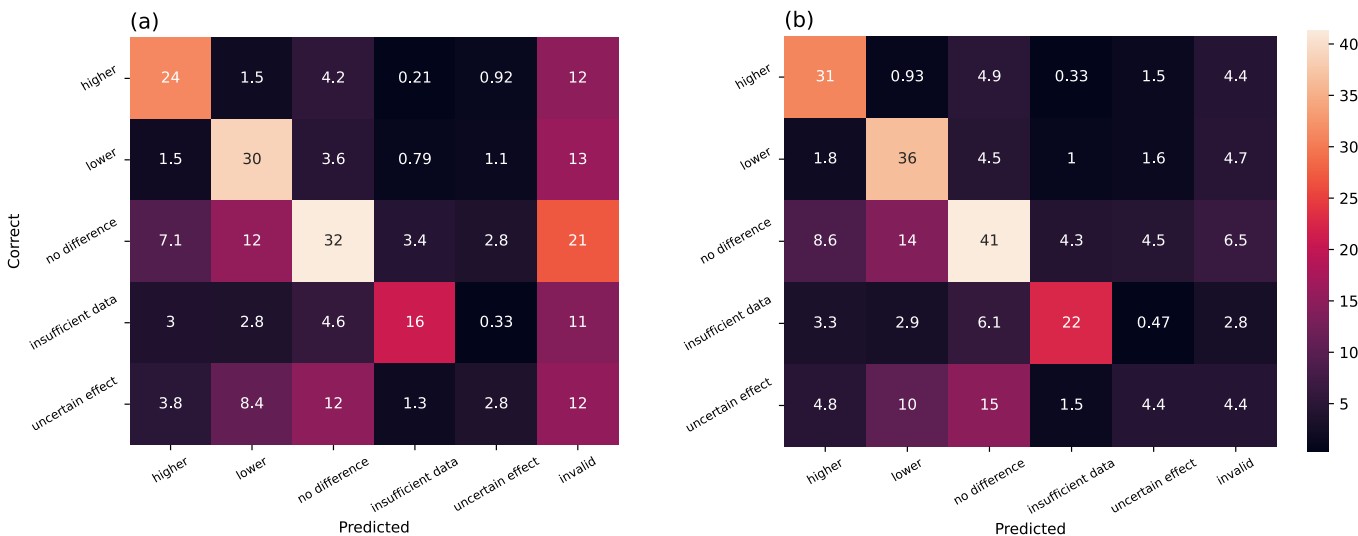

Figure 17: Average confusion matrices using basic prompts. (a) Average confusion matrix aggregated across all models. (b) Average confusion matrix aggregated across models achieving at least 40% overall accuracy.

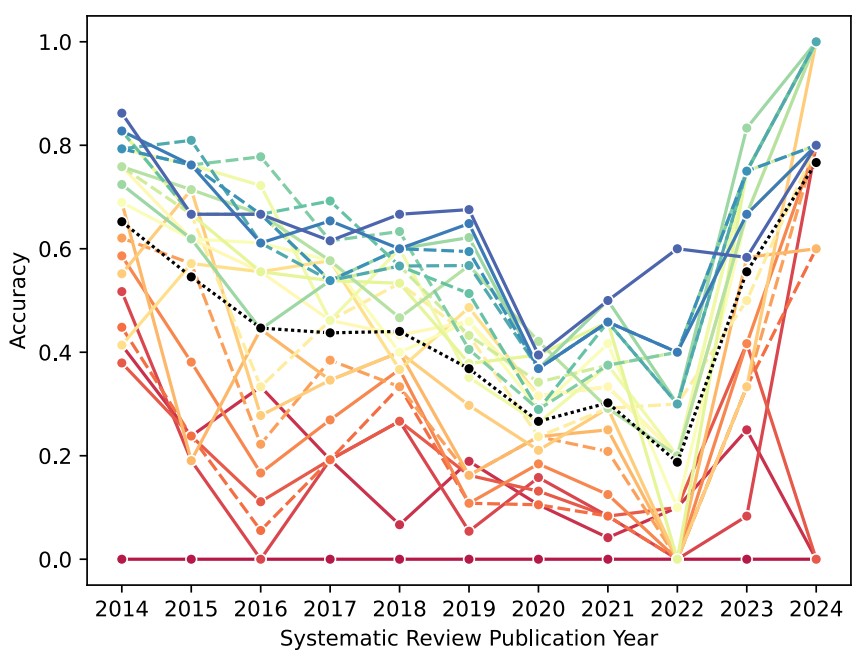

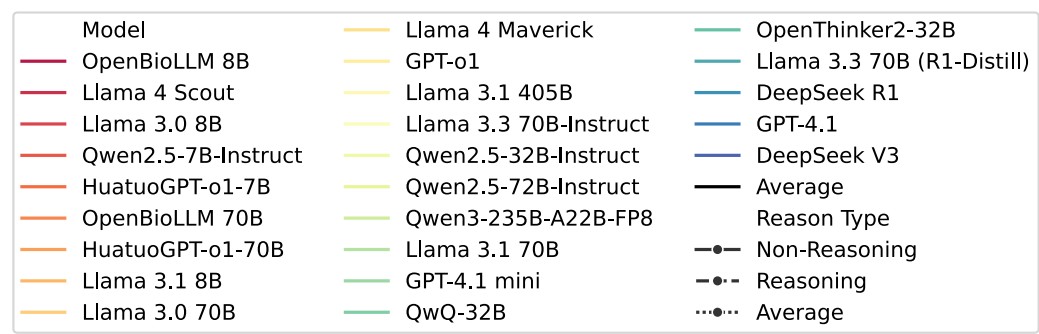

Figure 18: Accuracy by publication year

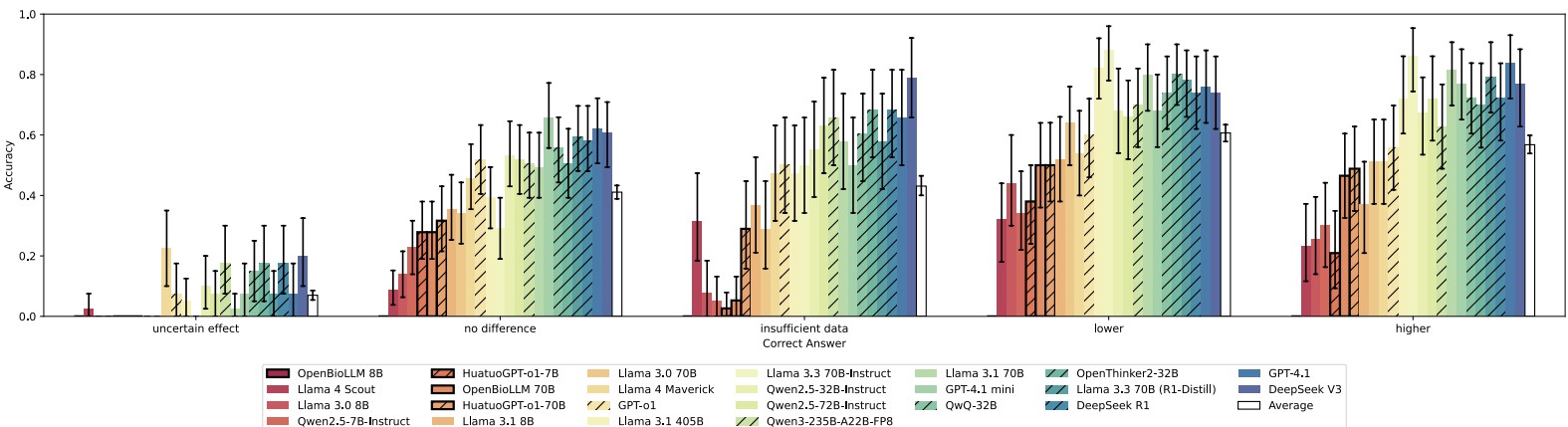

Figure 19: Per-class recall for each individual model. 95% confidence intervals are calculated via bootstrapping with N=1000.

## K   Model performance under the expert-guided prompt setup

To evaluate the dependency of model performance on prompting quality, we leverage an expert-guided prompt setup as described in the main paper and Appendix Section D. Critically, as shown in Appendix Figure 20 and discussed in the main paper, we find that even with a prompt explicitly designed to encourage models to assess the quality of studies, the dependency of model performance on evidence certainty remains. More broadly, as shown in Appendix Figure 21, we find that our more intentionally-designed prompt does not consistently improve model performance; while performance improves for the five models that performed worst under the basic prompt (namely OpenBioLLM 8B, Llama 4 Scout, Llama 3.0 8B, Qwen2.5-7B-Instruct, and HuatuoGPT-o1 7B), we observe that performance actually decreases for several of the models that performed best with the basic prompt, including a nearly 20% drop in performance for DeepSeek V3 (the highest-performing model when using the basic prompt).

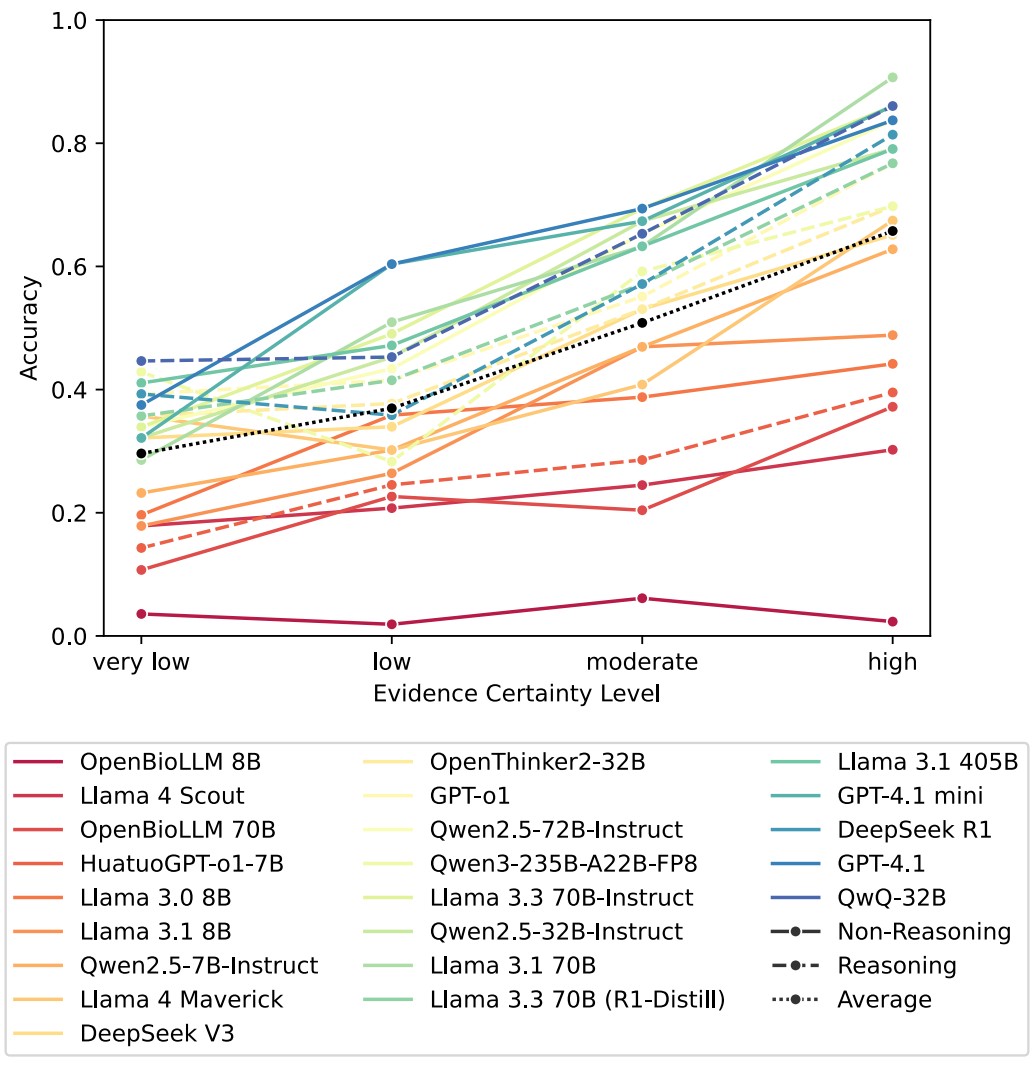

Figure 20: Model accuracy at different evidence qualities when using the expert-guided prompt setup. HuatuoGPT-o1 70B and Llama 3.0 70B are omitted as they were not tested on the expert-guided setup.

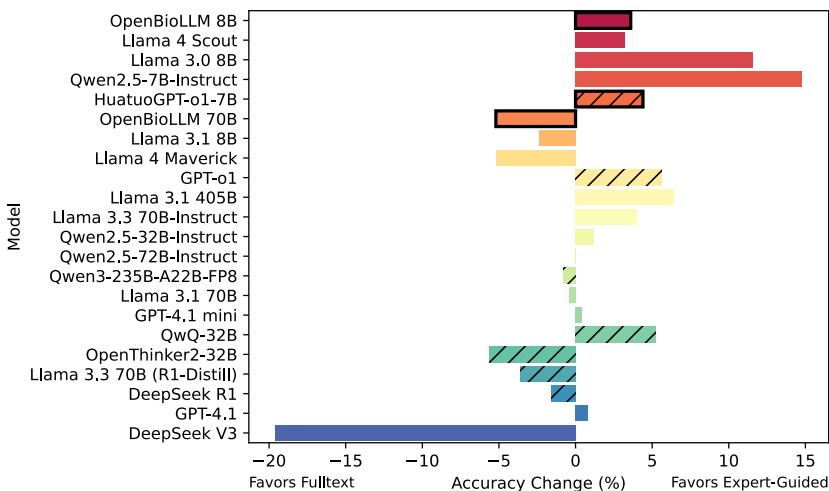

Figure 21: Changes in model performance when using the basic prompt setup versus the expert-guided prompt setup. HuatuoGPT-o1 70B and Llama 3.0 70B are omitted as they were not tested on the expert-guided setup.

## L Question correctness across models

As shown in Appendix Figure 22, 53 questions are answered incorrectly by all models, and only 2 are answered correctly by all models (omitting OpenBioLLM 8B, which gets every question wrong). Otherwise, we observe that performance varies significantly across models . A qualitative analysis of these various question types is presented in Appendix Section O.

## M Performance by medical specialty

Appendix Figure 23 shows average model accuracy stratified by medical specialty. Models perform significantly worse on questions relating to Psychology & Neurology and Surgery relative to other medical specialties, with accuracies of 27.60% (24.58, 30.52) and 34.09% (31.15, 37.03) respectively. The highest average model performance is observed in the Oncology & Hematology specialty, where models achieve an average accuracy of 63.28% (95% CI: 58.33–68.23).

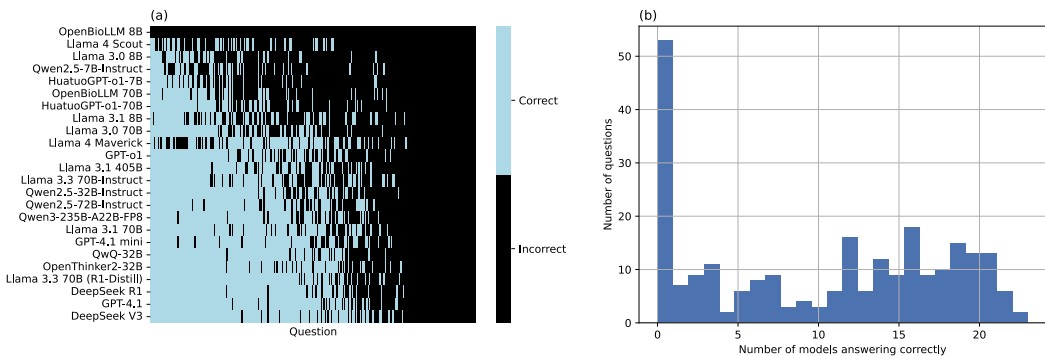

Figure 22: Analyses of model behavior across questions. (a) Questions (columns) that were deemed correct (light blue) or incorrect (black) for each model (rows), sorted by percentage of models with correct responses for that question (x-axis) and by the percentage of questions a model got correct (y-axis). (b) Distribution of questions by the number of models that answered that question correctly.

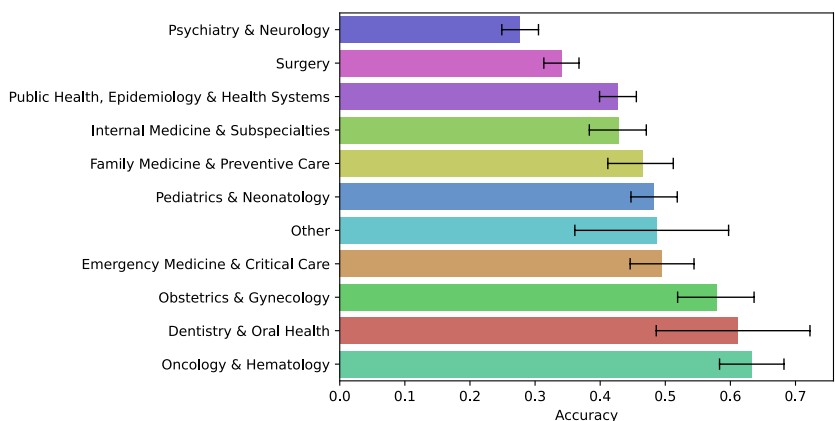

Figure 23: Average model accuracy across all models (and 95% confidence interval) stratified by medical specialty.

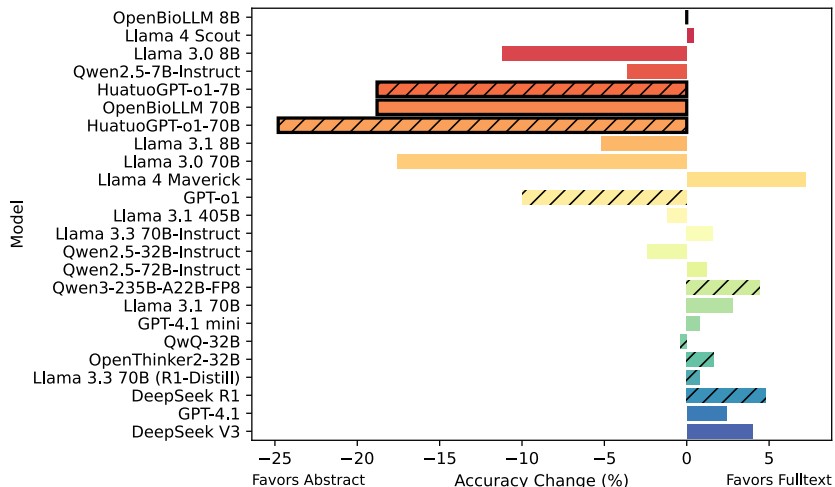

Figure 24: Changes in model performance when providing full-text when available versus always providing only the abstract (basic prompt setup).

# N Full-text vs abstract sources

We evaluate how model performance differs when using full-text articles versus abstracts alone, using the basic prompt setup in both cases. The results of this experiment are presented in Appendix Figure 24. We find that most models with the poorest overall performance actually experience a boost in accuracy (as high as 24.8% in the case of HuatuoGPT-o1-70B) when given only abstracts, even though abstracts contain less information. This suggests that some models struggle on our dataset because of an inability to handle long context, as full-text articles are much longer than abstracts alone. By contrast, the best-performing models usually perform better when given access to full-text (DeepSeek R1, for instance, gains 4.8% accuracy with full-text access). We note that, due to licensing constraints in scientific publishing, many existing deployments and evaluations of LLM to answer questions using scientific literature rely solely on abstracts [2, 7]. Our analysis demonstrates that access to full article text benefits frontier models, underscoring the urgent need to expand such access. We highlight initiatives promoting this shift—for instance, beginning in 2025, all U.S. federally-funded research must be made freely available, which could significantly enhance the performance of already-deployed models.

## O   Qualitative analysis of DeepSeek V3

To provide a better qualitative understanding of success and failure modes on our dataset, we explore the outputs (under the basic prompt setup) of our best-performing model, DeepSeek V3, and provide some commentary on the quality of the model's output. Specifically, we consider four types of questions: (1) Questions where all models get the answer wrong. (2) Questions where most models are correct, but DeepSeek V3 is incorrect. (3) Questions where most models are correct, including DeepSeek V3. (4) Questions where most models are wrong, but DeepSeek V3 is correct.

### O.1   Questions where all models are wrong

---

**Question ID 1 - 0/24 models correct**

**Review PMID:** 28660687
**Question:** Is the short-term rate of overall lymphocyst formation higher, lower, or the same when comparing retroperitoneal drainage to no drainage?
**Correct Answer:** `no difference`
**PMIDs of Relevant Sources:** 9190979, 12214830

---

**DeepSeek V3 Response:**

```
- **Rationale**:
  - Article ID 9190979 reports that the complication rate, including
     lymphocyst formation, was significantly higher in the drained group (
     Group A) compared to the not-drained group (Group B) (P = 0.01).
     However, sonographic monitoring showed free abdominal fluid (a proxy
     for lymphocyst formation) in 18% of drained patients vs. 36% of not-
     drained patients (P = 0.03), suggesting a lower rate in the drained
     group. This creates a contradiction in the findings.
  - Article ID 12214830 found no significant difference in lymphocyst
     formation between the drained and not-drained groups (P = 0.2), with
     asymptomatic lymphocysts detected only in the not-drained group (though
      statistically insignificant).

- **Full Answer**:
  - Article ID 9190979 reports a higher complication rate (including
     lymphocyst-related morbidity) in the drained group [9190979], but
     sonographic data suggests a lower rate of free abdominal fluid (a proxy
      for lymphocyst formation) in the drained group (18% vs. 36%, P = 0.03)
      [9190979]. Article ID 12214830 found no significant difference in
     lymphocyst formation between groups (P = 0.2) [12214830]. The evidence
     is conflicting, with one study suggesting a lower rate and another
     showing no difference.

- **Answer**: Uncertain Effect
```

---

**Remarks:** In the SR, authors perform meta-analysis and find no overall significant difference between retroperitoneal drainage and no drainage. DeepSeek V3 does not appear to know how to perform any kind of reconcilliation. We note that Article 12214830 does not specify the raw number of events under both treatments, but does state that no lymphocysts (i.e. 0) were detected in the drainage group.

---

**Question ID 198 - 0/24 models correct**

**Review PMID:** 35514111
**Question:** Is the rate of any PCR-positive SARS-COV-2 infection higher, lower, or the same when comparing test-based attendance to standard 10-day self-isolation?
**Correct Answer:** `uncertain effect`
**PMIDs of Relevant Sources:** 34534517

---

**DeepSeek V3 Response:**

- **Rationale**: The study compared daily testing (test-based attendance)
  with standard 10-day self-isolation for school-based contacts of SARS-CoV
  -2 cases. The primary outcomes included rates of symptomatic PCR-
  confirmed infections and COVID-19-related absences. The results showed no
  significant difference in the rate of symptomatic PCR-confirmed
  infections between the two groups, with the intervention group (daily
  testing) being non-inferior to the control group (self-isolation). The
  adjusted incidence rate ratio (aIRR) for symptomatic PCR-confirmed
  infections was 0.96 (95% CI -0.751.22), indicating no statistically
  significant difference.

- **Full Answer**: The rate of PCR-positive SARS-CoV-2 infection was not
  significantly different when comparing test-based attendance (daily
  testing) to standard 10-day self-isolation (aIRR 0.96, 95% CI -0.751.22)
  [34534517].

- **Answer**: No Difference

**Remarks:** The authors of the SR state that the results are inconclusive for the following reasons: (a) "risk of bias: considerable missing data for PCR-tests for staff (data were available for 76% control schools and 83% intervention schools)" and (b) "imprecision: the confidence intervals are wide and the 95% CI crosses the line of no effect." In this case, the full-text article is open-source, so LLMs theoretically had access to essentially all the same information as the SR authors, but evidently, none of these concerns are addressed in DeepSeek V3's response. Furthermore, the DeepSeek V3 response focuses only on symptomatic PCR-confirmed infections, demonstrating limitations in its knowledge of medical contexts, as the question calls for an assessment of any PCR positive infections, which includes asymptomatic cases.

970

---

## Question ID 32 - 0/24 models correct

**Review PMID:** 26447539
**Question:** Is subjective fatigue higher, lower, or the same when comparing methylphenidate to placebo?
**Correct Answer:** no difference
**PMIDs of Relevant Sources:** 17674415

**DeepSeek V3 Response:**

- **Rationale**: The study (Article ID 17674415) directly compares
  methylphenidate to placebo in treating fatigue in Parkinson's disease
  patients. The results show that methylphenidate significantly reduced
  fatigue scores (FSS and MFI) compared to placebo, where reductions were
  nonsignificant. This indicates that subjective fatigue is lower with
  methylphenidate than with placebo.

- **Full Answer**: Subjective fatigue is lower with methylphenidate compared
  to placebo, as evidenced by significant reductions in Fatigue Severity
  Scale (FSS) and Multidimensional Fatigue Inventory (MFI) scores in the
  methylphenidate group (P < 0.04), while placebo group reductions were
  nonsignificant [17674415].

- **Answer**: Lower

**Remarks:** This is a relatively challenging question—in the original SR, the authors compare the improvements in FSS in the placebo group directly to the improvements in the treatment group, rather than comparing the two treatment groups separately to their own baselines

971

(which is what the authors of the cited source do). The source's abstract does state that smaller but non-significant improvements were present in the placebo group, albeit without providing numbers. Regardless, the LLM's response reads as a regurgitation of facts and lacks the critical thinking needed to critically assess cited studies.

## O.2 Questions where most models are correct, but DeepSeek V3 is wrong

### Question ID 6 - 16/24 models correct

**Review PMID:** 26346232
**Question:** Is stroke prevention higher, lower, or the same when comparing Transcatheter Device Closure (TDC) to medical therapy?
**Correct Answer:** no difference
**PMIDs of Relevant Sources:** 22417252, 23514285, 23514286

---

**DeepSeek V3 Response:**

```
- **Rationale**: The three articles provide mixed evidence on the
    effectiveness of Transcatheter Device Closure (TDC) compared to medical
    therapy for stroke prevention in patients with patent foramen ovale (PFO)
    . Article ID 22417252 found no significant difference in stroke
    prevention between TDC and medical therapy (P=0.37 for the primary
    endpoint). Article ID 23514285 also reported no significant reduction in
    recurrent embolic events or death with TDC (P=0.34 for the primary
    endpoint). However, Article ID 23514286 showed no significant benefit in
    the intention-to-treat analysis (P=0.08) but did find a significant
    reduction in recurrent stroke in the per-protocol and as-treated analyses
     (P=0.03 and P=0.007, respectively). Given the inconsistency in results
    across studies and analyses, the overall evidence is not conclusive.

- **Full Answer**: The evidence is mixed. Two studies (Article ID 22417252
    and Article ID 23514285) found no significant difference in stroke
    prevention between TDC and medical therapy [22417252][23514285]. A third
    study (Article ID 23514286) reported no significant benefit in the
    intention-to-treat analysis but found TDC superior in per-protocol and as
    -treated analyses [23514286].

- **Answer**: Uncertain Effect
```

---

**Remarks:** In this case, unlike most models (16/24 to be exact), DeepSeek V3 seems to 'overthink' the question and hone in on a single case where a sub-analysis of in Article 23514286 finds a significant benefit. Notably, SRs typically use intention-to-treat analysis (https://training.cochrane.org/handbook/current/chapter-06), which DeepSeek V3 fails to consider. Furthermore, as in prevous cases, no clear efforts to perform meta-analysis or aggregate studies are shown in DeepSeek V3's response.

### Question ID 106 - 15/24 models correct

**Review PMID:** 38197546
**Question:** Is the risk of cholera at two-year follow-up higher, lower, or the same when comparing two doses of Dukoral with or without a booster dose to placebo?
**Correct Answer:** lower
**PMIDs of Relevant Sources:** 7967990, 10823767

---

**DeepSeek V3 Response:**

```
- **Rationale**:
```

- Article ID 7967990 evaluates the short-term efficacy of two doses of WC/
  rBS cholera vaccine without a booster, showing 86% protective efficacy
  over a mean follow-up of 18 weeks. However, it does not provide data on
  long-term (two-year) efficacy or the effect of a booster dose.
- Article ID 10823767 evaluates the two-year efficacy of the same vaccine,
  comparing two doses (without booster) to three doses (with booster).
  After two doses, the vaccine showed no efficacy (VE = -4%), but after
  the booster dose, efficacy increased to 61%. This suggests that the
  booster dose significantly improves long-term protection compared to
  two doses alone.

- **Full Answer**:
  - Without a booster dose (two doses only), the vaccine showed no protective
    efficacy (VE = -4%) at two-year follow-up [10823767].
  - With a booster dose (three doses), the vaccine demonstrated 61%
    protective efficacy at two-year follow-up [10823767].
  - Therefore, the risk of cholera at two-year follow-up is **higher** with
    two doses alone compared to two doses plus a booster.

- **Answer**: Higher

---

**Remarks:** This is case where DeepSeek V3 seems to fail due to a misinterpretation of the question, although we believe the semantics of this question are unambiguous. DeepSeek V3 does not seem to pick up on the mention of placebo in the question at all.

---

## Question ID 163 - 16/24 models correct

**Review PMID:** 25739381
**Question:** Is neonatal mortality higher, lower, or the same when comparing community-led chlorhexidine cord cleansing to community-led dry cord care?
**Correct Answer:** lower
**PMIDs of Relevant Sources:** 22322124, 16546539, 22322126

---

**DeepSeek V3 Response:**

- **Rationale**: The question asks whether neonatal mortality is higher,
  lower, or the same when comparing community-led chlorhexidine cord
  cleansing to community-led dry cord care. The three articles provide
  evidence from randomized trials in different settings (Bangladesh, Nepal,
  and Pakistan) comparing chlorhexidine cord cleansing with dry cord care.
  All three studies report reductions in neonatal mortality with
  chlorhexidine use, though the magnitude and statistical significance vary
  .

- **Full Answer**:
  - In Bangladesh, neonatal mortality was lower with single chlorhexidine
    cleansing (22.5 per 1000 livebirths) compared to dry cord care (28.3
    per 1000 livebirths; RR 0.80 [0.65-0.98]) [22322124].
  - In Nepal, neonatal mortality was 24% lower with chlorhexidine (RR 0.76
    [0.55-1.04]) compared to dry cord care, with a significant 34%
    reduction in infants enrolled within 24 hours (RR 0.66 [0.46-0.95])
    [16546539].
  - In Pakistan, chlorhexidine cleansing reduced neonatal mortality (RR 0.62
    [0.45-0.85]) compared to dry cord care [22322126].

- **Answer**: **Lower**

---

**Remarks:** An example of an instruction-following failure for DeepSeekV3 (even though the overall analysis leads to a correct solution).

 **O.3 Questions where most models are correct, including DeepSeek V3**

---

### Question ID 160 - 21/24 models correct

**Review PMID:** 28829911
**Question:** Is the risk of treatment-related mortality higher, lower, or the same when comparing chemoradiotherapy plus esophagectomy to chemoradiotherapy alone?
**Correct Answer:** higher
**PMIDs of Relevant Sources:** 17401004, 15800321

---

**DeepSeek V3 Response:**

```
- **Rationale**: Both articles compare chemoradiotherapy plus esophagectomy (
    surgery) to chemoradiotherapy alone in patients with locally advanced
    esophageal cancer. Article ID 17401004 reports a 3-month mortality rate
    of 9.3% in the surgery group (arm A) compared to 0.8% in the
    chemoradiation-alone group (arm B), with a statistically significant
    difference (P = .002). Article ID 15800321 similarly reports a
    significantly higher treatment-related mortality in the surgery group
    (12.8%) compared to the chemoradiotherapy-alone group (3.5%) (P = .03).
    Both studies consistently demonstrate that adding surgery to
    chemoradiotherapy increases treatment-related mortality.

- **Full Answer**: The risk of treatment-related mortality is higher when
    comparing chemoradiotherapy plus esophagectomy to chemoradiotherapy alone
    , as evidenced by significantly higher mortality rates in the surgery
    groups (9.3% vs. 0.8% [17401004]; 12.8% vs. 3.5% [15800321]).

- **Answer**: Higher
```

---

**Remarks:** While this question involves multiple sources, both sources individually agree with the overall correct conclusion, making it fairly straightforward to answer; consequently, almost all models get this question correct.

---

### Question ID 214 - 23/24 models correct

**Review PMID:** 37594020
**Question:** Is pain relief at 48 hours higher, lower, or the same when comparing NSAIDs to placebo?
**Correct Answer:** higher
**PMIDs of Relevant Sources:** 8871138

---

**DeepSeek V3 Response:**

```
- **Rationale**: The study compared ibuprofen (an NSAID) to acetaminophen and
     placebo for pain relief in children with acute otitis media over 48
    hours. The results showed that ibuprofen was statistically superior to
    placebo (P < 0.01) in relieving pain, with 7% of children still suffering
     pain in the ibuprofen group compared to 25% in the placebo group. There
    was no significant difference between acetaminophen and placebo for pain
    relief.

- **Full Answer**: Pain relief at 48 hours was higher with NSAIDs (ibuprofen)
     compared to placebo, as evidenced by a statistically significant
    reduction in pain (7% vs. 25% of children still suffering pain, P < 0.01)
     [8871138].

- **Answer**: Higher
```

---

**Remarks:** A straightforward question that all models (except OpenBioLLM 8B) get correct:

one source accurately directly reports on relevant findings without conclusions being revised by the SR authors' critical analysis.

## O.4 Questions where DeepSeek V3 is correct, despite most models being wrong

---

### Question ID 235 - 3/24 models correct

**Review PMID:** 32104914
**Question:** Is quality of life higher, lower, or the same when comparing computerised cognitive training (CCT) of at least 12 weeks to no intervention?
**Correct Answer:** `insufficient data`
**PMIDs of Relevant Sources:** 23531885, 25511081, 20418350

---

**DeepSeek V3 Response:**

```
- **Rationale**: The question asks whether quality of life (QoL) is higher,
    lower, or the same when comparing computerized cognitive training (CCT)
    of at least 12 weeks to no intervention. The provided articles focus on
    cognitive and physical performance improvements but do not explicitly
    measure or report QoL outcomes. Article ID 23531885 discusses cognitive
    improvements from CCT but does not mention QoL. Article ID 25511081
    examines dual-task gait and divided attention but also does not address
    QoL. Article ID 20418350 evaluates cognitive performance after mental and
     physical activity interventions but similarly lacks QoL data.

- **Full Answer**: None of the articles provide data on quality of life (QoL)
     outcomes when comparing CCT of at least 12 weeks to no intervention
    [23531885, 25511081, 20418350].

- **Answer**: Insufficient Data
```

---

**Remarks:** DeepSeek V3 correctly recognizes that none of the relevant studies provide direct, quantitative analysis on QoL outcomes; given the very low model correctness rate (3/24), other models likely attempted to apply colloquial definitions of "quality of life" rather than the narrowly-defined medical definition requiring explicit quantitative measurement.

---

### Question ID 89 - 2/24 models correct

**Review PMID:** 31557310
**Question:** Is the incidence of surgical site infection (SSI) for patients undergoing breast cancer surgery without reconstruction higher, lower, or the same when comparing prophylactic antibiotics administered preoperatively to placebo?
**Correct Answer:** `lower`
**PMIDs of Relevant Sources:** 7480237, 9776150, 23052809, 10873356, 19673597, 2403655, 2403697

---

**DeepSeek V3 Response:**

```
- **Rationale**: The question asks about the incidence of surgical site
    infection (SSI) in breast cancer surgery without reconstruction when
    comparing prophylactic antibiotics to placebo. The relevant articles
    provide mixed results: some show a significant reduction in SSI with
    antibiotics (e.g., [7480237], [2403655]), while others show no
    significant difference (e.g., [23052809], [10873356], [2403697]). Notably
    , [7480237] specifically highlights a significant reduction in infections
     for breast surgery, while [2403655] also supports a reduction in
    infections for breast surgery. However, other studies, such as [23052809]
     and [10873356], found no significant difference. The variability in
```

results suggests that the effect may depend on the type of breast surgery
    (e.g., mastectomy vs. axillary lymph node dissection) or other factors.

- **Full Answer**: The incidence of SSI is **lower** with prophylactic
    antibiotics compared to placebo in some breast cancer surgeries, as shown
    by significant reductions in infections in studies involving plastic
    surgery ([7480237]) and certain breast procedures ([2403655]). However,
    other studies found no significant difference for modified radical
    mastectomy ([23052809]), clean elective breast surgery ([10873356]), or
    mastectomy ([2403697]). The overall evidence suggests a context-dependent
    effect, with some procedures benefiting from prophylaxis.

- **Answer**: Lower

---

**Remarks:** While DeepSeek V3 does get this question right, and shows flashes of attempting deeper reasoning (e.g. the remark that the effect "may depend on the type of breast surgery"), but this analysis is not directly supported by similar statements from the original SR authors. Additionally, there is still no formal aggregation or meta-analysis attempted, and the level of detail in the reasoning provided by DeepSeek V3 makes it hard to tell if the model's predicted answer was made via some form of intuition or if the model just got a bit lucky.

## Question ID 173 - 1/24 models correct

**Review PMID:** 31425604
**Question:** Is the risk of necrotizing enterocolitis higher, lower, or the same when comparing monitoring of gastric residual volume and quality to only monitoring of gastric residual quality?
**Correct Answer:** uncertain effect
**PMIDs of Relevant Sources:** 29866595

---

**DeepSeek V3 Response:**

- **Rationale**: The study compared two groups: one where gastric residual
    volume (GRV) was measured only in the presence of specific clinical signs
    (study group) and another where GRV was routinely measured (control
    group). The primary outcome was time to reach full feeding volumes, but
    secondary outcomes included necrotizing enterocolitis (NEC). The study
    found no significant differences in NEC incidence between the two groups,
    with only 2 cases reported in the control group and none in the study
    group. However, the small sample size (87 infants) and low event rate
    limit the ability to draw definitive conclusions about NEC risk.

- **Full Answer**: The study found no significant difference in the incidence
    of necrotizing enterocolitis between the group where gastric residual
    volume was measured only in the presence of clinical signs (0 cases) and
    the group where it was routinely measured (2 cases) [29866595]. The small
    sample size and low event rate suggest uncertainty in the effect.

- **Answer**: Uncertain Effect

---

**Remarks:** Even without explicit prompting, DeepSeek V3 recognizes the weakness of the limited sample size/total number of events—the fact that only DeepSeek V3 gets this question correct shows both the current limitations of models' ability to assess uncertainty, as well as the promise that they may be able to do so consistently in the future.

## P   Individual confusion matrices for all models

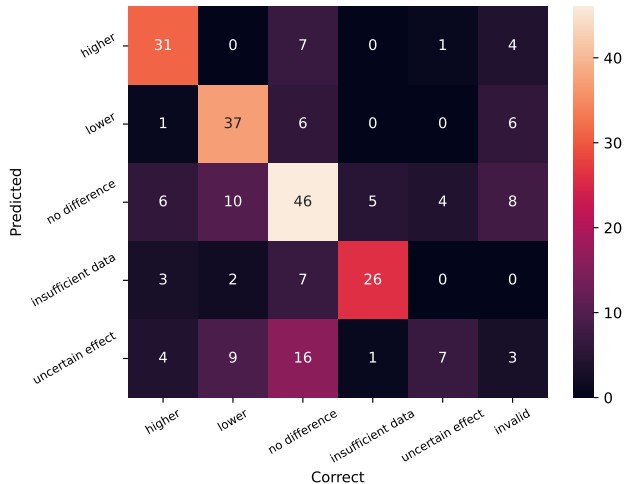

Figure 25: Confusion matrix for DeepSeek R1.

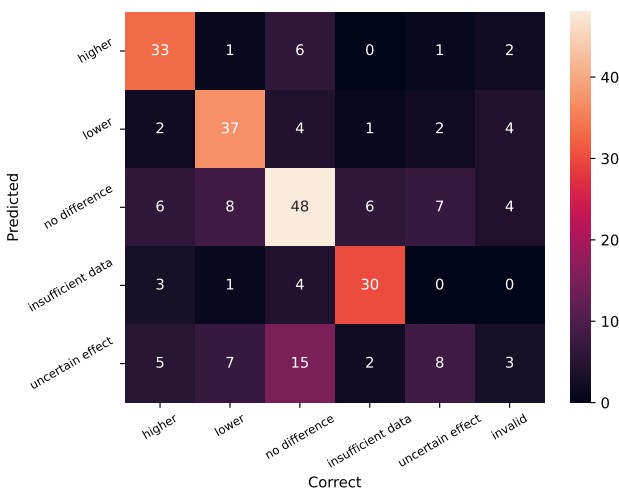

Figure 26: Confusion matrix for DeepSeek V3.

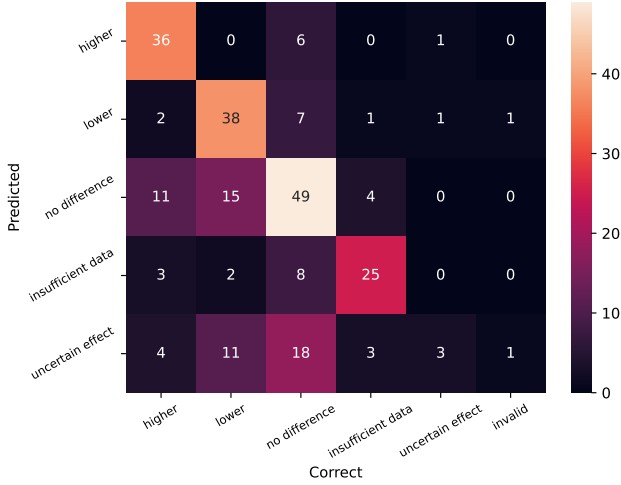

Figure 27: Confusion matrix for GPT-4.1.

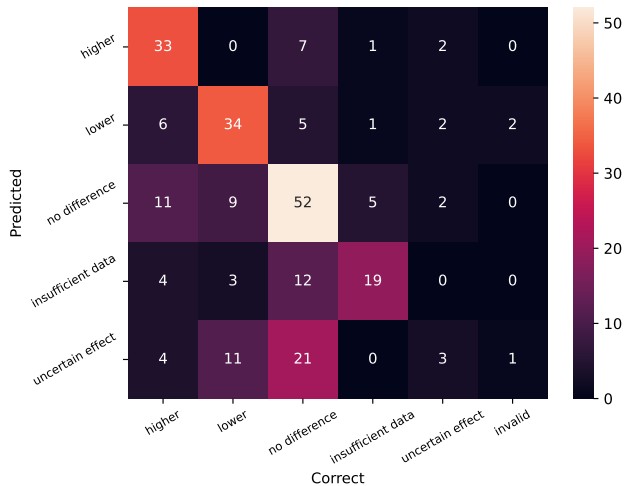

Figure 28: Confusion matrix for GPT-4.1 mini.

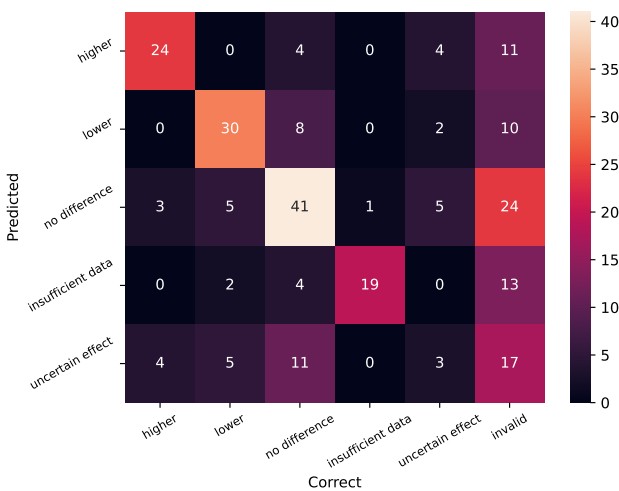

Figure 29: Confusion matrix for GPT-o1.

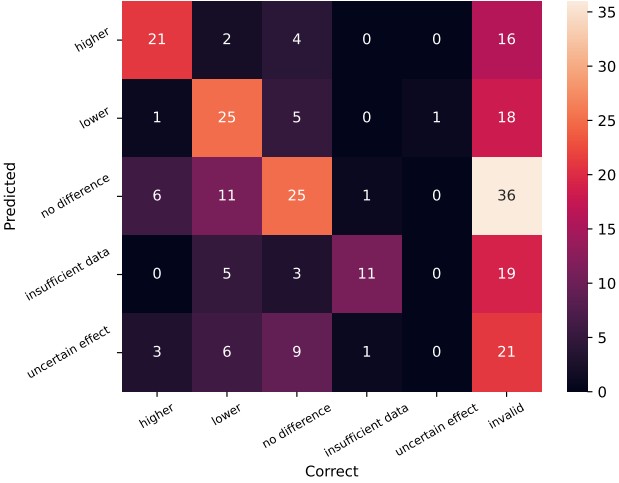

Figure 30: Confusion matrix for HuatuoGPT-o1-70B.

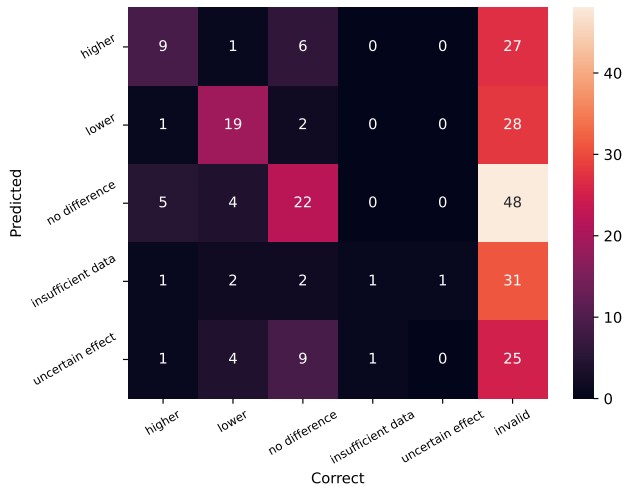

Figure 31: Confusion matrix for HuatuoGPT-o1-7B.

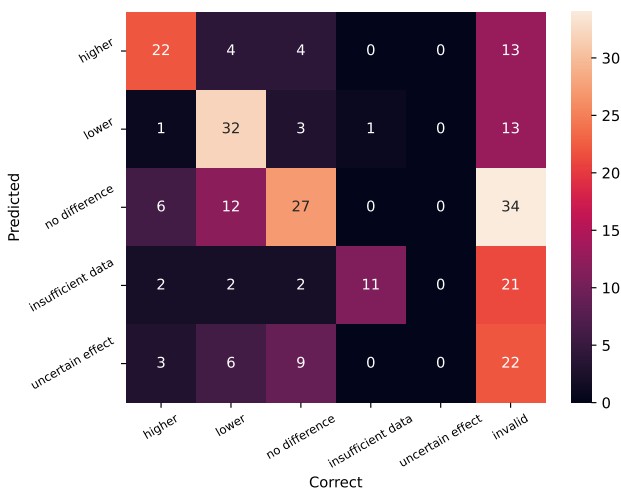

Figure 32: Confusion matrix for Llama 3.0 70B.

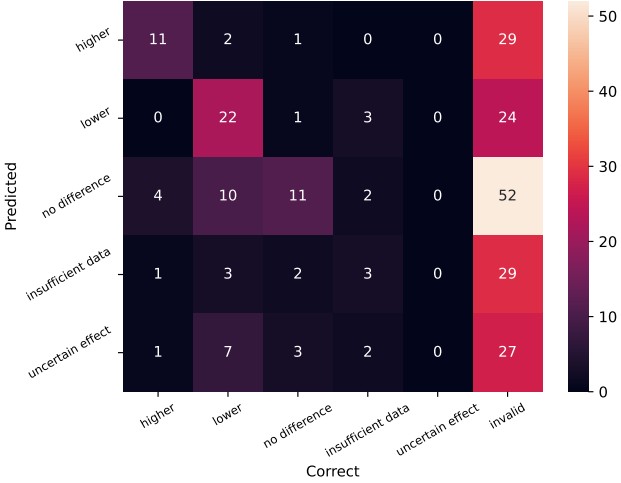

Figure 33: Confusion matrix for Llama 3.0 8B.

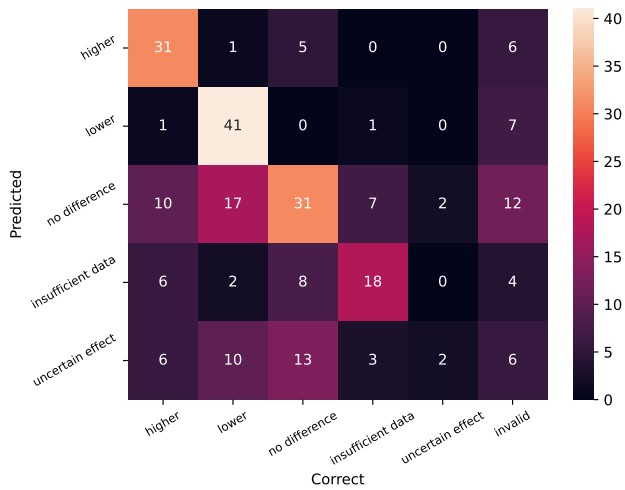

Figure 34: Confusion matrix for Llama 3.1 405B.

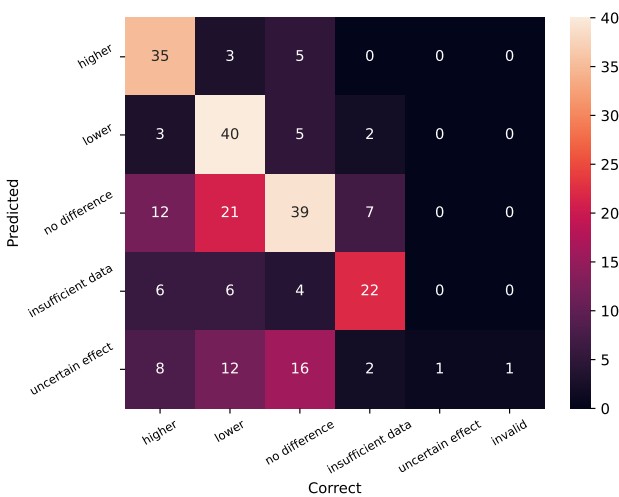

Figure 35: Confusion matrix for Llama 3.1 70B.

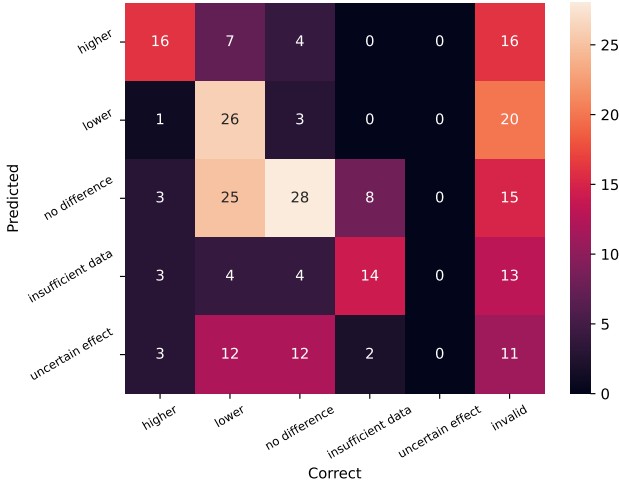

Figure 36: Confusion matrix for Llama 3.1 8B.

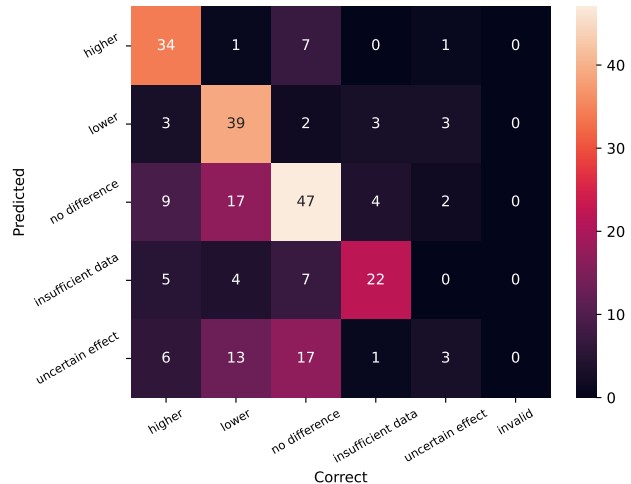

Figure 37: Confusion matrix for Llama 3.3 70B (R1-Distill).

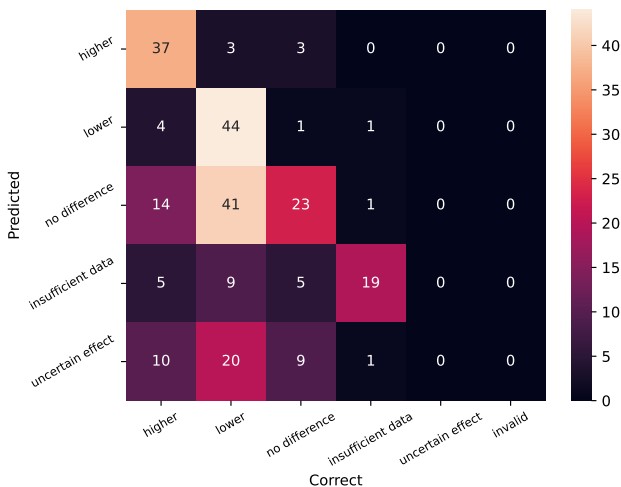

Figure 38: Confusion matrix for Llama 3.3 70B-Instruct.

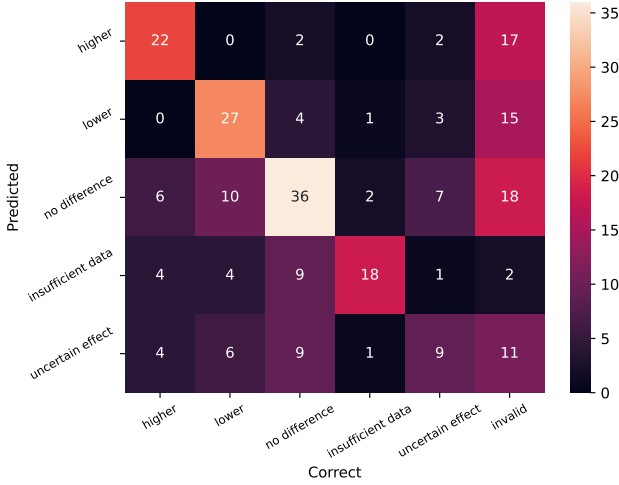

Figure 39: Confusion matrix for Llama 4 Maverick.

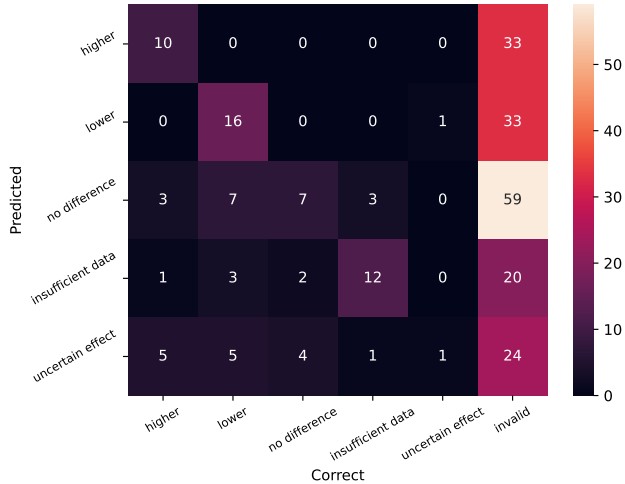

Figure 40: Confusion matrix for Llama 4 Scout.

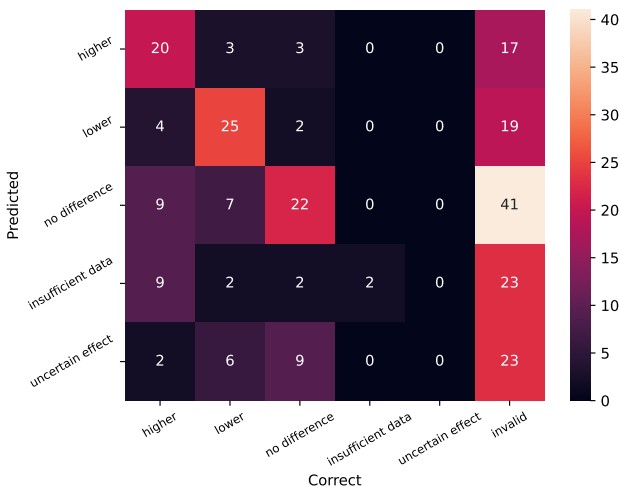

Figure 41: Confusion matrix for OpenBioLLM 70B.

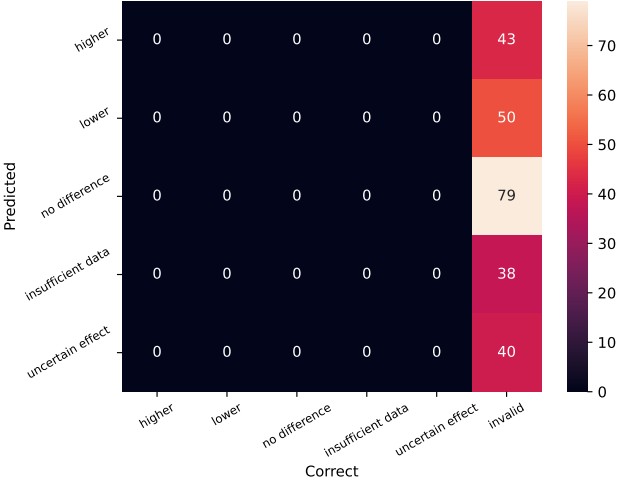

Figure 42: Confusion matrix for OpenBioLLM 8B.

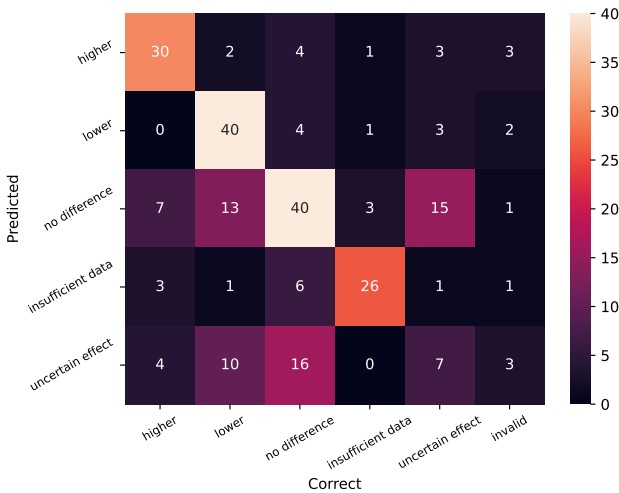

Figure 43: Confusion matrix for OpenThinker2-32B.

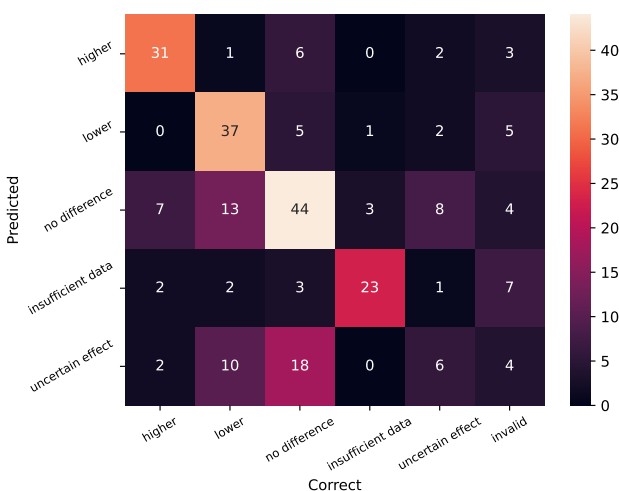

Figure 44: Confusion matrix for QwQ-32B.

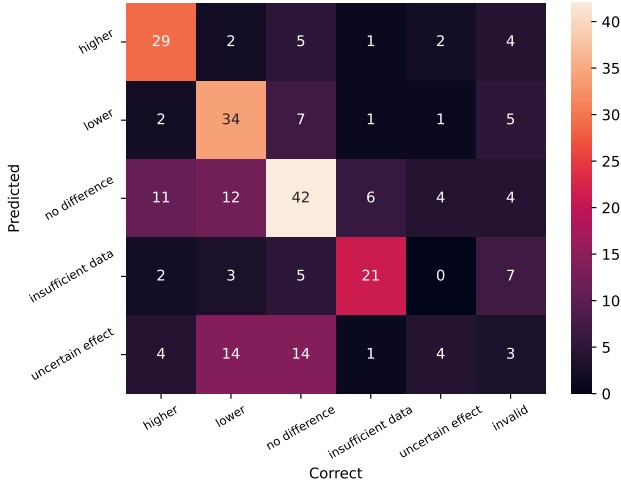

Figure 45: Confusion matrix for Qwen2.5-32B-Instruct.

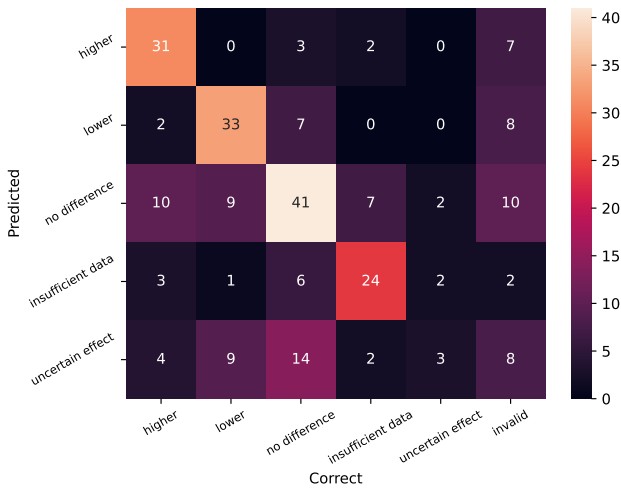

Figure 46: Confusion matrix for Qwen2.5-72B-Instruct.

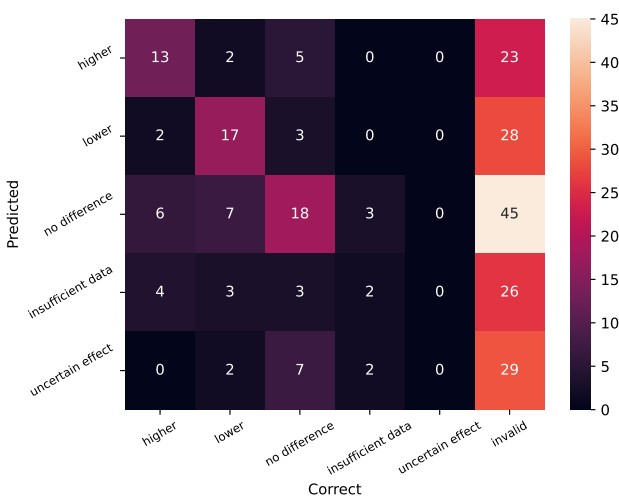

Figure 47: Confusion matrix for Qwen2.5-7B-Instruct.

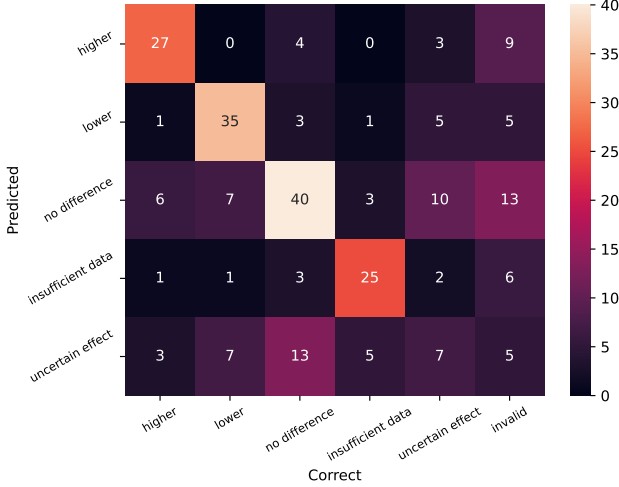

Figure 48: Confusion matrix for Qwen3-235B-A22B-FP8.