# OpenReview forum: "Can LLMs Match the Observations of Systematic Reviews?"
_NeurIPS.cc/2025/Datasets_and_Benchmarks_Track — Submitted to NeurIPS 2025 Datasets and Benchmarks Track_

### Official Review · Reviewer_H2VM · 2025-06-01

**Rating:** 4
**Confidence:** 5

**Summary:**

This paper introduces MedEvidence, a benchmark for evaluating whether large language models (LLMs) can reproduce the conclusions of clinical systematic reviews (SRs) when given access to the same source studies. The authors curate 284 question–answer pairs from 100 open‐access Cochrane SRs spanning ten medical specialties. Each question asks, for example, “Is outcome X higher, lower, or the same when comparing intervention A to control B?” Human annotators (graduate‐level) extract the relevant studies, evidence certainty (GRADE), and full‐text requirements. The LLMs are prompted in zero‐shot fashion (with optional “expert‐guided” framing) to provide both a rationale and a single answer from five classes (higher/lower/no difference/uncertain/insufficient data). The paper benchmarks 24 diverse LLMs—ranging from 7 B to 671 B parameters, including “reasoning” variants and medically fine‐tuned models—using accuracy (exact match) and analyzes performance across key factors: token‐length, evidence certainty, source concordance, model size, and fine‐tuning. The main finding is that even state‐of‐the‐art LLMs (e.g., GPT‑4.1, DeepSeek V3) achieve only ~60–62% accuracy (well below human experts) and exhibit overconfidence, poor skepticism toward low‐quality evidence, and diminishing returns beyond 70 B parameters.

**Dataset Code Accessibility:**

Yes

**Ethical Considerations:**

No, there are no or only very minor ethics concerns

**Limitations Weaknesses:**

Dataset Size Is Modest (284 Questions, 100 SRs)

Although each question is carefully curated, 284 items may still be insufficient to fully capture the breadth of clinical SR conclusions. Many specialties (e.g., oncology, cardiology) routinely publish dozens of SRs per year. The limited sample invites possible sampling bias: 100 SRs may not reflect the full diversity of clinical questions.

Suggestion: Expand to 500–1,000 questions covering additional SRs, especially in underrepresented areas (e.g., dermatology, endocrinology). A larger scale could better stabilize accuracy estimates and permit fine‑tuning of smaller open‑source models.

Dependence on Abstracts When Full‑Text Is Unavailable

For ~215 of the 329 referenced articles, only abstracts were retrieved (due to full‑text paywalls). Abstracts often omit subgroup analyses, secondary endpoints, or nuanced statistical caveats. Consequently, some “insufficient data” cases might be misclassified because the abstract lacked key details.

Suggestion: In future revisions, either (a) restrict the benchmark to questions whose all sources have open‑access full text or (b) explicitly label questions supported by abstracts only, so that models need not “hallucinate” details outside the abstract.

Fixed Five‐Way Answer Format Limits Nuance

The categorical answer set {“higher”, “lower”, “no difference”, “uncertain effect”, “insufficient data”} covers the majority of SR conclusions, but real SRs also include qualifiers such as “low‐certainty evidence suggests…” or “possible but not clinically significant difference.” Collapsing everything into five buckets may oversimplify nuanced language.

Suggestion: Add a sixth category, e.g., “clinically insignificant,” or allow a free‐text “explanation” field, so that answers can reflect the difference between “statistically significant but clinically marginal” and “truly no difference.”

Prompt Engineering Limited to Zero‑Shot and One “Expert‑Guided” Template

The paper compares only two prompting strategies: a minimal zero‑shot prompt (rationale + single‐phrase answer) and an “expert‑guided” prompt asking the model to grade evidence. However, it is known that chain‑of‑thought (CoT) prompting or few‑shot exemplar injection can significantly improve multi‑document reasoning. This evaluation may understate what LLMs could achieve if given a handful of annotated examples or explicit CoT triggers.

Suggestion: Include at least one few‑shot / CoT prompt variant (e.g., provide two fully annotated QA examples before the test question) to assess whether LLMs can leverage demonstrations. Reporting these results would clarify whether the bottleneck is the model’s reasoning architecture or simply the lack of guidance.

Lack of Human‑Baseline Performance

While the paper describes that “human experts” originally generated each SR conclusion, it never reports a direct human performance on MedEvidence using only abstracts/full text (as the LLMs see). It would be informative to have 2–3 clinicians manually answer a random subset of 50 questions (with time constraints matched to an LLM’s inference time) to quantify the human overshoot: e.g., “experts achieve 85% accuracy (CI ±3%),” showing that the gap between LLMs (≈60%) and humans (≈85%) is concrete.

Suggestion: Add a small human‐baseline experiment on at least 50 randomly sampled questions, measuring inter‐annotator agreement and average accuracy.

**Strengths Contributions:**

Novel, Controlled Benchmark for SR Conclusions

MedEvidence is, to the authors’ knowledge, the first publicly released dataset that (a) pairs SR conclusions with the exact source studies used by experts, (b) converts each conclusion into a closed‐form QA item, and (c) provides metadata for evidence certainty (GRADE) and source concordance. Unlike earlier factuality or medical‐QA benchmarks that test internal LLM knowledge or single‐document fact verification, MedEvidence simulates the genuine SR reasoning process: synthesizing multiple sources with varying quality and sometimes conflicting results. This fills a clear gap in “multi‐source evidence synthesis” evaluation.

Diverse, High‑Quality Data Curation

The authors curate 284 questions from 100 Cochrane SRs (2014–2024), ensuring each is grounded in peer‐reviewed meta‐analyses across 10 specialties (e.g., surgery, psychiatry, pediatrics). Three graduate‐level annotators extract “Main Results” statements, convert them into the five‐way QA format, select relevant studies via SR appendices, and validate that ≥75% of “evidence weight” comes from source abstracts or full text. Evidence certainty is directly harvested from each SR’s GRADE assessment. This careful, multi‑stage pipeline (selection → conversion → study selection → feasibility validation) minimizes noise and yields a dataset that closely mirrors expert practice.

---

> ### Author Rebuttal · Authors · 2025-07-31
>
> Thank you for your careful evaluation of our manuscript, and for supporting our dataset as a meaningful contribution to the community. We are motivated by your assessment that our dataset is "diverse" and "high-quality" and "fills a clear gap in 'multi‐source evidence synthesis' evaluation." Below, we address your suggestions for additional improvements:
>
> **R4.1 ”Dataset Size Is Modest (284 Questions, 100 SRs). Although each question is carefully curated, 284 items may still be insufficient to fully capture the breadth of clinical SR conclusions. [...] Suggestion: Expand to 500–1,000 questions covering additional SRs, especially in underrepresented areas (e.g., dermatology, endocrinology). ”**
>
> We understand your concern. We devoted substantial effort to exploring automated pipelines for dataset generation, but ultimately found them infeasible. As a result, we opted for a manual QA conversion process to ensure quality, at the cost of a smaller sample size, and high annotation time: 100 hours of annotations (4.1 days)  spread through a month to avoid annotation burnout and ensure consistent quality.
>
> While the dataset is modest in size, this is not without precedent; other evaluation datasets, such as ConflictingQA and ClinicalQA (Almanac), also contain fewer than 300 questions. Given the complexity of our task—including full-text comprehension, conflicting source resolution, and critical assessment—we believe the dataset is a first step to benchmark meaningful future progress in LLM-assisted systematic reviews.
>
> Our experience curating the dataset suggests that significantly expanding it—as proposed by the reviewer and desired by us—risks disrupting its balance and could lead to misleading model evaluations. During curation, we addressed several challenges, in particular, SRs citing papers without available abstracts, abstracts lacking statistical summaries of treatment-outcome effects, and meta-analyses based on only 1–2 studies. Based on our current pipeline, we estimate that ~60-70% of the high-quality questions have already been captured. Expanding to 500+ questions would likely lead to a sharp rise in questions with only a single relevant source, a reduction in cases requiring full-text comprehension, or a need to lower our quality threshold for abstracts. Balancing underrepresented areas would also be difficult, as we have limited control over the availability and quality of SR references.
>
> However, we do agree that a more expansive dataset would provide additional benefits to the community, and hope that actions like the NIH mandate to make NIH-founded research open-source will make the scalability of our dataset a reality in the years to come.
>
> **R4.2 ”Add a sixth category, e.g., “clinically insignificant,” or allow a free‐text “explanation” field, so that answers can reflect the difference between “statistically significant but clinically marginal” and “truly no difference.”**
>
> We acknowledge that omitting a "clinically insignificant" category limits our ability to fully assess models' understanding. However, in the SRs used to construct the dataset, authors did not consistently distinguish between "no difference" and "clinically marginal" effects. To avoid introducing our own bias when interpreting these nuances, we chose not to add a sixth category.
>
> That said, we do provide two free-text fields ("Rationale" and "Full Answer") that allow models to elaborate. As shown in Appendix O, and further discussed in our response to R3.3, our qualitative inspection suggests that while models occasionally recognize the concept of clinical significance, they rarely demonstrate the ability to assess it reliably on their own.
>
> **R4.3 “Include at least one few‑shot / CoT prompt variant (e.g., provide two fully annotated QA examples before the test question) to assess whether LLMs can leverage demonstrations. Reporting these results would clarify whether the bottleneck is the model’s reasoning architecture or simply the lack of guidance."**
>
> Thank you for pointing this out. By default, all our prompts include CoT prompting. Based on the recommendations of several reviewers, we have added a few-shot and CoT analysis to the paper—please see R3.3 for more details regarding few-shot and CoT performance (TL;DR - few-shot learning does improve performance of some models, but the general trends and upper limit of performance remain the same; prompting with or without CoT results in statistically-equivalent performance).
>
> **Actions:** Added few-shot and CoT analyses (please see R3.3).
>
> **R.4.4 “Add a small human‐baseline experiment on at least 50 randomly sampled questions, measuring inter‐annotator agreement and average accuracy.”**
>
> Thank you for your suggestion; we agree that a human-baseline is valuable. Due to time constraints, we recruited three external clinicians to answer 20 randomly selected questions using the same inputs provided to the language models—please see R1.4 for more details (TL;DR - the mean accuracy was 71.67%; every question was answered correctly by at least one clinician; inter-rater agreement was Fleiss' kappa of 0.43).
>
> **Actions:** Added a human-baseline experiment (please see R1.4 for details).
>
> Thank you once again for your thoughtful review and insightful commentary, which have helped us further improve the manuscript.  Please don’t hesitate to reach out with any additional questions or comments!

---

> ### Author Response · Authors · 2025-08-06
>
> Thank you again for your initial feedback on our manuscript. We are reaching out again to ensure that we have addressed your concerns. We are happy to clarify further if needed!

---

### Official Review · Reviewer_b8AT · 2025-06-02

**Rating:** 5
**Confidence:** 5

**Summary:**

This paper introduces MedEvidence, a novel benchmark evaluating large language models (LLMs) on their ability to replicate individual conclusions from expert-written medical systematic reviews, given access to the same source studies. The authors curate 284 questions from 100 Cochrane systematic reviews across 10 medical specialties, transforming review findings into closed-form question-answering tasks. The benchmark is used to evaluate 24 LLMs, revealing limitations in evidence synthesis, skepticism toward low-quality studies, and overconfidence. Surprisingly, models with medical fine-tuning often underperform their base counterparts. These findings highlight key limitations of current LLMs in replicating expert-level medical evidence synthesis.

**Dataset Code Accessibility:**

Yes

**Dataset Code Comments:**

The code is available on GitHub, and the dataset is hosted on HuggingFace; both are linked in the paper and its OpenReview profile.

**Ethical Considerations:**

No, there are no or only very minor ethics concerns

**Final Justification:**

Considering the original quality of the paper and the authors’ clarifications during the discussion, I decide to maintain my original score. While some concerns remain, the core contribution is solid.

**Limitations Weaknesses:**

1. Limited exploration of prompting techniques: Only two prompting strategies (basic and expert-guided) are tested. The paper could better explore how in-context examples or retrieval-based chain-of-thought prompting might improve performance, especially on “uncertain” questions.
2. Evaluation restricted to one SR subtask: The benchmark evaluates only whether models can match SR conclusions. Other SR components (study screening, bias evaluation, narrative synthesis) are excluded. While acknowledged as a limitation (Section 6), this reduces the generalizability of findings to full SR pipelines.
3. Minimal qualitative analysis of model errors: The failure analysis is primarily aggregate (e.g., per-class recall, Fig. 4). A deeper dive into why models fail (e.g., hallucination, anchoring bias, reliance on recent study) would strengthen insights for future work.

**Strengths Contributions:**

1. Timely and impactful benchmark: MedEvidence addresses a critical and practical challenge—whether LLMs can match expert-level systematic review conclusions. Given the growing adoption of LLMs in clinical tools, this is a highly relevant contribution.
2. High-quality dataset curation: The benchmark is carefully constructed using Cochrane SRs, a trusted and standardized source. The four-stage annotation pipeline (systematic review selection, QA transformation, source matching, feasibility validation) is robust and well-documented.
3. Meaningful evaluation framework: Unlike prior benchmarks (e.g., MedREQAL, ConflictingQA), MedEvidence uses the original source evidence rather than pre-synthesized claims, which better approximates real systematic review tasks. The comparison in Table 1 clearly shows its uniqueness.
4. Comprehensive LLM evaluation: The paper systematically evaluates 24 models across size, reasoning capability, and fine-tuning type. Key trends such as degradation with long inputs, ineffectiveness of medical fine-tuning (Fig. 6), and model overconfidence are convincingly supported. The analysis also reveals deeper behavioral patterns—such as reluctance to express uncertainty, performance tied to source concordance, and lack of skepticism toward weak evidence—making the benchmark a useful tool for probing reasoning limitations.
5. Reproducibility and accessibility: Dataset and code are publicly released, with clear experiment settings and statistical reporting.

---

> ### Author Rebuttal · Authors · 2025-07-31
>
> We sincerely appreciate your thorough review and positive feedback on our submission. We are encouraged that you find our work “timely and impactful” given the “high-quality dataset curation” that enables “reproducibility and accessibility,” and thank you for highlighting our “comprehensive LLM evaluation” and “meaningful evaluation framework.” Below, we address your suggestions for further refinement:
>
> **R3.1 "Limited exploration of prompting techniques: Only two prompting strategies (basic and expert-guided) are tested [...] explore how in-context examples or [...] chain-of-thought prompting might improve performance”**
>
> Thank you for your comment. We clarify that our existing prompts all include a “think step by step” prompt at the end (as shown in Appendix D). However, to further address this concern, we have added two experiments, as described below.
>
> **Actions:**
> - **Few-shot learning experiment**: We held out 3 examples showcasing 3 non-trivial interpretations of the evidence, then evaluated several models on all remaining questions using 1-3 of the examples. We highlight that given the large context required to run such an experiment, we report performance on models with long-context support (context window of at least 131K tokens). As shown in the table below, adding in-context examples significantly improves the performance of some weaker models (notably Llama 4 Maverick), but does not raise accuracy beyond the previously observed accuracy ceiling in the low 60s. It also has minimal impact on top-performing zero-shot models, such as DeepSeek V3 and Llama 3.3 70B R1-distill. Overall, our findings remain consistent: error rates are similar across models and setups, zero-shot DeepSeek V3 remains one of the best models, and a performance gap between LLMs and human experts persists. The large improvements for Llama 4 Maverick can be at least partially explained by a large jump from around 60% valid output to over 95% valid output (where valid means the model followed instructions, as defined in Appendix E).
>
> | Model | 0-shot | 1-shot | 2-shot | 3-shot |
> |:--------------------------|--------:|:---------------|:---------------|:---------------|
> | Llama 4 Maverick | 0.448 | 0.583 (+0.135) | 0.583 (+0.135) | 0.632 (+0.184) |
> | Llama 3.3 70B-Instruct | 0.492 | 0.555 (+0.063) | 0.571 (+0.079) | 0.579 (+0.087) |
> | GPT-4.1 mini | 0.564 | 0.619 (+0.055) | 0.595 (+0.031) | 0.619 (+0.055) |
> | Llama 3.3 70B (R1-Distill) | 0.58 | 0.599 (+0.019) | 0.591 (+0.011) | 0.591 (+0.011) |
> | DeepSeek V3 | 0.624 | 0.567 (-0.057) | 0.555 (-0.069) | 0.575 (-0.049) |
>
> - **CoT omission experiment**: To evaluate the importance of chain-of-thought prompting, we reran our baseline experiment on several models with the "Think step-by-step" removed from the instruction prompt. Based on this experiment, we find that omitting or including CoT does not result in any statistically-significant difference.
>
> | Model                      |   With CoT |   No CoT |   Delta |
> |:---------------------------|-----------:|---------:|--------:|
> | Llama 3.0 70B              |      0.368 |    0.36  |   0.008 |
> | Qwen2.5-72B-Instruct       |      0.528 |    0.512 |   0.016 |
> | GPT-4.1 mini               |      0.564 |    0.564 |   0     |
> | Llama 3.3 70B (R1-Distill) |      0.58  |    0.576 |   0.004 |
> | DeepSeek V3                |      0.624 |    0.604 |   0.02  |
>
> **R3.2  “Evaluation restricted to one SR subtask: The benchmark evaluates only whether models can match SR conclusions. Other SR components (study screening, bias evaluation, narrative synthesis) are excluded. While acknowledged as a limitation (Section 6), this reduces the generalizability of findings to full SR pipelines.”**
>
> We appreciate this observation and acknowledge the limitation. Our decision to focus solely on the synthesis step is intentional, motivated by our core research question: *Can LLMs match the conclusions of systematic reviews when given the same set of studies as human experts?* This isolates the LLM’s ability to interpret, integrate, and reason over clinical evidence—particularly in the presence of conflicting findings, which remains an underexplored capability.
>
> By removing the document retrieval step, typically handled by external tools like BM25 or dense retrievers, we avoid introducing confounding variables. For instance, if a retriever returns irrelevant documents, the LLM’s performance would be penalized for factors unrelated to synthesis ability. Our setup ensures a fairer evaluation of the LLM’s core reasoning over a curated evidence base.
>
> **Actions:** We agree that assessing the entire pipeline of systematic reviews is valuable. However, given the substantial annotation effort involved (~100 hours), we chose to begin with the synthesis step. We have revised the introduction to clarify this motivation and expanded our discussion to underscore that our benchmark does not evaluate end-to-end review generation. Rather, it focuses on whether LLMs can produce similar conclusions to experts when operating over the same source material.
>
> **R3.3 “Minimal qualitative analysis of model errors: The failure analysis is primarily aggregate (e.g., per-class recall, Fig. 4). A deeper dive into why models fail (e.g., hallucination, anchoring bias, reliance on recent study) would strengthen insights for future work.”**
>
> We agree that qualitative analysis is imperative to complement quantitative findings. We highlight that while we originally provided a qualitative analysis in Appendix O, we did not make use of it in the discussion. To this end, we have added a small subsection in the discussion section to discuss qualitative findings.
>
> Additionally, while our original exploration only includes the best model in our benchmark (DeepSeek V3), we have expanded the qualitative analysis to the best three performing models and the worst three performing models, and added this expanded qualitative analysis to Appendix O. To summarize the findings: we observe a few key failure modes in all models: **failures to reconcile conflicting evidence**; **failures to critically assess sources** (especially recognizing that study design or sample sizes are too small to draw definitive conclusions); and **inaccurate understanding of medical terminology** (leading to models believing irrelevant claims in the text are actually relevant to the question).
>
> The best-performing models (DeepSeekV3, GPT4.1, and DeepSeekR1) are more likely to reconcile the conflicting evidence correctly, although the mechanism by which they do this is not clearly stated in the output rationale, and they still often make mistakes. Meanwhile, the worst-performing models (OpenBioLLM-8B, Llama 4 Scout, and Llama 3.0 8B) suffer most heavily due to a consistent lack of instruction-following, and greater rates of medical terminology misinterpretation compared to the best models. In the case of Llama 3.0 8B, in addition to the failure modes of the better models, it also appears to sometimes misinterpret the evidence it cites when mapping its full answer to one of the answer classes.
>
> We note that many of these behaviors have examples in the existing Appendix O.
>
> Thank you again for your thoughtful review and insightful comments to help us improve our manuscript further. We believe these revisions enhance the clarity of our contributions. Please feel free to reach out if you have any further questions or comments. We would be happy to discuss them further!

---

> > ### Comment · Reviewer_b8AT · 2025-08-08
> >
> > Thank you for your detailed response. Several of my concerns have been addressed.

---

### Official Review · Reviewer_bvug · 2025-06-12

**Rating:** 2
**Confidence:** 4

**Summary:**

This paper introduces MedEvidence, a benchmark that evaluates LLMs' abilities to replicate the conclusions extracted from expert-written systematic reviews when provided with the abstract or full-text of the source studies from which these conclusions were derived. The benchmark contains 284 manually-curated questions from 100 Cochrane reviews. It analyses the performance of GPT-o1 and a range of open-source models (both general and fine-tuned on the biomedical domain) and finds that accuracy is modest even for frontier models.

**Additional Feedback:**

- I also wonder whether it is meaningful to require models to perform implicit meta-analysis, which could be easily achieved with instructions and tool-use.
- There doesn't seem to be an agreed upon definition of what constitutes an "uncertain effect". This puts the usability of ground-truth answers in this category into question.

**Dataset Code Accessibility:**

Yes

**Ethical Considerations:**

No, there are no or only very minor ethics concerns

**Final Justification:**

I appreciate the discussion with the authors and some of the additional analyses they performed.

My core methodological concern remains: the **class labels are non-standard and derived from subjective wording** rather than objective criteria based on effect sizes, uncertainty, quality of evidence, etc. Concrete benchmark questions like ID 1 and ID 3 show that these can be inconsistent or ambiguous, making error attribution difficult. The additional small experiments, in particular a human baseline and a re-evaluation without the "uncertain effect" class) are informative but remain compatible with these issues and do not resolve them.

For a benchmark paper, **trust in label validity and reproducibility is essential**; without a robust labeling, I cannot recommend acceptance.

**Limitations Weaknesses:**

I have considerable doubts about the quality of the curated questions. Just looking at the first four questions (IDs 0-3) sourced from two Cochrane reviews, I found several issues:
- The answer provided in the ground-truth review text ("no difference") doesn't always match the underlying meta-analysis results, which showed considerable heterogeneity (completely oppsite effects of the 2 included studies, 95%-CI for risk ratio ranging from 0.04 to 13.35, Tau2 = 3.38, I2 = 74.55%) und should be interpreted as "uncertain effect" (ID 1).
- The answer listed in the benchmark ("uncertain effect") doesn't always match the the answer provided by the meta-analysis results and the review text ("no difference"; ID 3).
- The LLMs are essentially asked to perform implicit meta-analyses but are only provided with the study abstracts, which often only contained p-values rather than the information that would actually be necessary to perform the required meta-analysis (IDs 1-3).

Having found severe issues in 3 out of the first 4 questions, I have great doubts about any of the results presented in the paper.

**Strengths Contributions:**

The study addresses a narrow but important question. Using Cochrane reviews is a good approach. Examples were manually curated.

---

> ### Author Rebuttal · Authors · 2025-07-31
>
> Thank you for your careful inspection of the construction of our dataset, and for validating both the importance of our task and our use of Cochrane reviews as source material. We appreciate you raising your concerns, as they underscore the inherent difficulty in interpreting these questions and building the benchmark. Given these challenges, we chose to construct every question manually (approximately 20 minutes per question; see Appendix B.1) to minimize the risk of hallucinations from language models. Below, we address your concerns with the dataset construction:
>
> **R2.1 “I have considerable doubts about the quality of the curated questions. Just looking at the first four questions (IDs 0-3) sourced from two Cochrane reviews, I found several issues[...]**
>
> While the meta-analysis is a crucial component, it does not provide the full picture; **expert interpretation** is also essential and was used to formulate the final conclusions of the SRs.
>
> **Our question-answer pairs are derived directly from the authors' comprehensive analysis and align with their stated interpretations. These authors often include clinical experts, systematic review methodologists, and statisticians - providing a gold standard in clinical practice.** Therefore, while we understand the source of confusion, we believe the claims in question are not valid. Below, we provide detailed evidence demonstrating that the statements in our dataset are aligned with the SR authors' words as intended.
>
> **R2.1.1** "The answer provided in the ground-truth review text ("no difference") doesn't always match the underlying meta-analysis results, which showed considerable heterogeneity (completely oppsite effects of the 2 included studies, 95%-CI for risk ratio ranging from 0.04 to 13.35, Tau2 = 3.38, I2 = 74.55%) und should be interpreted as "uncertain effect" **(ID 1)**."
>
> - For ID 1, the review text reads: "Regarding short‐term outcomes (within four weeks after surgery), retroperitoneal drainage was associated with *a comparable rate* of overall lymphocyst formation when all methods of pelvic peritoneum management were considered together." Thus, the phrasing of "a comparable rate" can be unambiguously mapped to the "No Difference" label.
>
> **R2.1.2** "The answer listed in the benchmark ("uncertain effect") doesn't always match the the answer provided by the meta-analysis results and the review text ("no difference"; **ID 3**)."
>
> - For ID 3, the review text reads: "However, the *treatment effect on other aspects of cognitive change were unclear*, measured by the Selective Reminding Test (3 studies, WMD 1.47, 95% CI ‐0.39 to 3.32, P = 0.12; high quality evidence), *patient's self‐reported impression of memory change* (2 studies, OR 1.67, 95% CI 0.93 to 3.00, P = 0.08; high quality evidence) [...]" The phrasing that the effects were "unclear" unambiguously interprets to the "Uncertain Effect" label.
>
> **R2.2 “The LLMs are essentially asked to perform implicit meta-analyses but are only provided with the study abstracts, [...]”**
>
> We offer several clarifications to address your concern:
> 1. As noted in the introduction, retrieval-augmented LLMs are already being used in clinical settings (e.g., OpenEvidence, Elicit). Our benchmark was designed to reflect this real-world usage. We evaluate models across multiple input configurations—abstract-only, full-text, and mixed. The expert-guided prompt setting further instructs models to consider study design, population, and includes a table with recommendation grades (Appendix Fig. 13).
> 2. Relying solely on abstracts is common in prior work (e.g., ClinfoAI, PubMedQA, BioASQ, Med-Gemini). To reduce this limitation, our dataset incorporates open-access full-texts where possible: 114 of 329 referenced articles are full-text, 127/284 questions include at least one full-text source, and 64 rely exclusively on full-text. We report model performance using abstracts vs. mixed sources (Appendix Fig. 23, Section N), showing that large models benefit from full-text, while smaller models degrade.
> 3. We conducted a human evaluation of 20 randomly selected questions with three external clinicians, providing only the information we use to evaluate the LLMs (please refer to R1.4). We observe that every question can be answered correctly by at least one clinician.
>
> **Actions:**
> - We have updated Fig. 1 to include human (clinician) performance.
> - We have calculated the performance of models when only leveraging full-text; we have added this analysis to the Appendix, and provide a summarized table below. Our analysis shows that models consistently perform better on questions that require abstracts only as compared to full-text only, demonstrating that current models actually perform worse when provided with the information that would be necessary to replicate full meta-analysis.
>
> | Model | Abstracts only | Fulltext only | Mixed sources |
> |:------------|------------:|----------:|-------------:|
> | DeepSeek V3 | 0.638 | 0.46 | 0.772 |
> | GPT-4.1 | 0.615 | 0.524 | 0.667 |
> | DeepSeek R1 | 0.6 | 0.524 | 0.632 |
> | Llama 3.3 70B (R1-Distill) | 0.638 | 0.429 | 0.614 |
> | OpenThinker2-32B | 0.608 | 0.46 | 0.614 |
> | QwQ-32B | 0.631 | 0.46 | 0.526 |
> | GPT-4.1 mini | 0.608 | 0.444 | 0.596 |
> | Llama 3.1 70B | 0.6 | 0.381 | 0.614 |
> | Qwen3-235B-A22B-FP8 | 0.562 | 0.46 | 0.561 |
> | Qwen2.5-72B-Instruct | 0.592 | 0.444 | 0.474 |
> | Qwen2.5-32B-Instruct | 0.608 | 0.413 | 0.439 |
> | Llama 3.3 70B-Instruct | 0.554 | 0.365 | 0.491 |
> | Llama 3.1 405B | 0.592 | 0.349 | 0.421 |
> | GPT-o1 | 0.631 | 0.302 | 0.281 |
> | Llama 4 Maverick | 0.377 | 0.46 | 0.596 |
> | Llama 3.0 70B | 0.577 | 0 | 0.298 |
> | Llama 3.1 8B | 0.377 | 0.333 | 0.246 |
> | HuatuoGPT-o1-70B | 0.6 | 0.032 | 0.035 |
> | OpenBioLLM 70B | 0.531 | 0 | 0 |
> | HuatuoGPT-o1-7B | 0.385 | 0.016 | 0 |
> | Qwen2.5-7B-Instruct | 0.269 | 0.063 | 0.193 |
> | Llama 3.0 8B | 0.338 | 0 | 0.053 |
> | Llama 4 Scout | 0.169 | 0.238 | 0.158 |
> | OpenBioLLM 8B | 0 | 0 | 0 |
> | Average | 0.504167 | 0.298208 | 0.386708 |
>
> **R2.3 “There doesn't seem to be an agreed upon definition of what constitutes an "uncertain effect". This puts the usability of ground-truth answers in this category into question.**
>
> Our definition of the "uncertain effect" class, which is based on SR author wording, is defined in Appendix B.2: "Conclusions where the authors expressed that uncertainty was too great to evaluate a treatment outcome effect were placed in the uncertain effect label class. Conclusions where authors assessed a difference, but then stated that they were very uncertain of their findings were deemed ambiguous and discarded." In short, the "uncertain effect" label was used when the authors explicitly stated that a treatment-outcome effect was "unclear" or "uncertain."
>
> We appreciate your constructive feedback. We believe that your valid concerns regarding data quality stems from a misunderstanding, which we have tried to clarify. Please let us know if you have any further concerns or questions.

---

> > ### Comment · Reviewer_bvug · 2025-08-03
> >
> > I thank the authors for their detailed reply. Unfortunately, they did not alleviate my main concerns and I have thus not updated my score. Please find a detailed justification below.
> >
> >
> > ### "No difference" vs "uncertain effect"
> >
> > There is [guidance by Cochrane EPOC](https://epoc.cochrane.org/sites/epoc.cochrane.org/files/uploads/Resources-for-authors2017/how_to_report_the_effects_of_an_intervention.pdf) on how statements of effect should be reported in a standardised manner based on the effect size and certainty of evidence . In particular:
> >
> > - "no difference" should only be ascertained if there is a **null finding with high certainty evidence**.
> > - "uncertain effect " should only be ascertained if there is **very low certainty evidence**.
> >
> > Applying this to the benchmark statements under question:
> >
> > In **ID 1**, the confidence interval for a rate ratio ranges from 0.04-13.35 with significant heterogeneity. No matter what the SR authors' words, this is about as low as certainty can be and the wording therefore goes directly **against** Cochrane's own guidance.
> >
> > In **ID 3**, the SR authors themselves rate the evidence as **high quality evidence**, making this clearly not an uncertain effect.
> >
> > Given this inconsistency of the benchmark's grund truth with official Cochrane  guidelines, it is no suprise to me that the authors find the following in Appendix H:
> >
> > > However, when considering exclusively models with above 40% performance, we observe two significant trends. First, models are consistently unwilling to predict uncertain effect. Second, models consistently confuse the uncertain effect and no difference classes.
> >
> > **Conclusion**: The benchmark would benefit from adopting a more well-defined classification scheme. Following the Cochrane EPOC guideline for standardised reporting of effects linked above could be an option, asking the models to **separately** interpret effect size and certainty of evidence.
> >
> >
> > ### Expert interpretation
> >
> > I agree with the authors that expert interpretation is essential. The authors state that reviews "often included clinical experts, systematic review methodologists, and statisticians - providing a gold standard". However, I am not convinced that the included studies automatically meet this high bar.
> >
> > To give a concrete example, ID 1 was written by two clinicians only, without any reported advice from systematic review methodologists or statisticians. It may therefore not come as a surprise that the review text contains a questionable statistical interpretation that goes against Cochrane's own guidance (see previous section) and led to it incorrectly being labelled as "no difference".
> >
> > Given the further absence of any stated experience in conducting high-quality systematic reviews, the study's three human annotators with "between one and five years of graduate education" also seem insufficient to catch such discrapencies.
> >
> > **Conlusion:** While Cochrane can serve as a powerful preselection of studies, collaboration with and annotation by dedicated SR experts seems highly advisable to make this meaningful benchmark.
> >
> >
> > ### Human comparison
> >
> > I thank the reviewer for performing a preliminary human comparison. Including a comprehensive human baseline seems to me to be the right direction to make the benchmark results interpretable.
> >
> > If I read R1.4 and the updated table above correctly, the human experts' maximum accuracy was 75% and thus **actually lower** than the 77.2% achieved by DeepSeek V3 on Mixed sources? This result to me reads quite different from the authors' conclusion that
> > > more work is still required before LLMs can reliably match the observations from expert-conducted SRs, even though these systems are already deployed and being used by clinicians.
> >
> > **Conlusion:** I strongly suggest to perform a larger, more structured, and more robust human evaluation, potentially with different groups of participants (e.g., practicing clinicans vs. systematic review methodologists)

---

> > > ### Author Response · Authors · 2025-08-04
> > > **Response (part 1)**
> > >
> > > Thank you for your continued clarification of your position and feedback on our work. We remind the reviewer that the authors' interpretation of the evidence depends not only on the statistical result of the meta-analysis, but also on the qualitative properties of the studies conducted, such as levels of evidence and potential biases assessed in the original papers that make up the meta-analysis.
> > >
> > > **R2: “In ID 1, [...] the wording therefore goes directly against Cochrane's own guidance.”**
> > >
> > > As stated by Cochrane Database of Systematic Reviews editorial policies: “As a minimum standard, every Cochrane review will be peer reviewed by at least one clinical/topic specialist and one statistician/methodologist” [1], all of whom follow several guidelines and protocols (including the one shared) to assess the rigor of each SR before it is accepted.  This critique contradicts the findings from two clinicians  with at least ~10 years of experience in the topic discussed, along with the peer reviewers. The article has been published for 13 years, accessed 2K times, and to our knowledge based on Cochrane retracted publication feature, has not been retracted.
> > >
> > > We therefore respectfully disagree with R2's assessment of the expert’s interpretation. In our view, challenging the original would require a complete re-analysis of the included studies, as was undertaken in the SR itself. Without this, we believe the conclusion of the Cochrane review remains the most rigorous and reliable interpretation available.
> > >
> > > **R2: ”In ID 3, the SR authors themselves rate the evidence as high quality evidence, making this clearly not an uncertain effect.”**
> > >
> > > We remind the reviewer that the quality of evidence is not the same as an author's conclusion and interpretation. Quality of evidence is a system designed to assess the degree of confidence in the accuracy of research findings. The “level of evidence pyramid" is an example resource we believe will help clarify this misunderstanding. The author's interpretation depends on statistical analysis, evidence quality, and specific study evaluation, not just evidence quality. Thus, as written by the authors (and quoted in the initial rebuttal), the effects are unclear, even though the evidence used to arrive at the conclusion is of high quality.
> > >
> > > We also reiterate that our study seeks to investigate whether "LLMs can match the conclusions [...] written by clinical experts," motivated by the real world usage of LLMs that mimics our evaluation scenario [2,3,4,5]. For all these reasons, we prioritize the wording of the SR authors, rather than standard guidelines. We hope this clarifies the confusion.
> > >
> > > **R2: “I am not convinced that the included studies automatically meet this high bar [...] ID 1 was written by two clinicians only [...] questionable statistical interpretation that goes against Cochrane's own guidance.”**
> > >
> > > We remind R2 of the Cochrane Database of Systematic Reviews editorial policies:  “As a minimum standard, every Cochrane review will be peer reviewed by at least one clinical/topic specialist and one statistician/methodologist” Thus, every SR in our dataset involves at least a topic-specialist clinician and statistical methodologist in the review process, plus the additional authors (experts of the specific topic). For example, in ID1, the authors have maintained the SR for around 10 years. We remind the reviewer that Cochrane SRs are "regarded as the “gold standard' for high‐quality information and are widely used to inform healthcare” [6].
> > >
> > > While it is possible to argue the quality of the SR, we believe it is necessary to use the appropriate tools (at the level or higher) of the original SR. Regardless, this is a critique of the data source (which the reviewer agreed was a "good approach" to use), not our method.
> > >
> > > **R2: “the further absence of any stated experience in conducting high-quality systematic reviews, the study's three human annotators [...] seem insufficient to catch such discrepancies."**
> > >
> > > We clarify that the elaboration of this manuscript included discussions with clinical experts (including those that have written SR) and professors of statistics. However, due to the substantial work required to perform an SR, even with the inclusion of experts, we still believe it is most appropriate to keep the original SR authors' interpretation.

---

> > > > ### Comment · Reviewer_bvug · 2025-08-06
> > > > **Reviewer response (part 1)**
> > > >
> > > > I appreciate the authors' continued engagement and will try to clarify my position. I am not criticising Cochrane as the data source, **I am criticising the annotation method** chosen by the authors.
> > > >
> > > > ### The problem with the annotations
> > > >
> > > > The validity of this benchmark rests on two main assumptions:
> > > >
> > > > 1. that Cochrane reviews are the gold standard and provide accurate systematic reviews and meta-analyses of the highest quality.
> > > > 2. that annotating examples solely on the basis of the wording (e.g., "a comparable rate" in **ID 1** or "unclear" in **ID 3**) is a valid and robust way to extract the conclusion of an SR.
> > > >
> > > > While I am open to accept Assumption 1 (although Cochrane is certainly not infallible), Assumption 2 is in my opinion very much up for debate.
> > > >
> > > > Let me illustrate the considerable problem of labelling the risk ratio of 0.76 (95% CI 0.04 to 13.35) in **ID 1** as "no difference". The rate of complication in the control group of the source studies was 17%. Applying the above CI, the SR results suggest that drainage (=the treatment) might lower the probability of complication to <1\%  or increase the probability of complication to >99%. Domain knowledge cannot compensate for the extreme statistical imprecision here.
> > > >
> > > > Now how could this phrasing make its way into a Cochrane review? If we take the full sentence written by the authors, we can see that the SR authors accompany their claim of "comparable rates" with both the estimated effect size and the quality of evidence:
> > > >
> > > > > retroperitoneal drainage was associated with a comparable rate of overall lymphocyst formation when all methods of pelvic peritoneum management were considered together **(2 studies; 204 women; RR 0.76, 95% CI 0.04 to 13.35; moderate‐quality evidence)**.
> > > >
> > > > These additional passages **provide crucial context** that the annotators' simple word matching ignores. While I'd continue to argue that the SR authors' choice of words was suboptimal, I do not think that they are problematic if given the entire sentence, let alone severe enough to be a cause for retraction. The statement can simply be read as a clinician's statement of "no significant effect". On the other hand, the authors' label of "no difference" is not unambiguous -- as the authors stated in their rebuttal -- but  problematic, at least as long as there is also an "uncertain effect" category.
> > > >
> > > > I also want to note that a better annotation strategy would **not** require a "complete re-analysis of the included studies", as the authors claim. All that would be needed are objective annotation criteria that do not rely on ambiguous wording taken out of context.
> > > >
> > > >
> > > > ### This issue extends into the higher/lower class
> > > >
> > > > Upon closer inspection of Appendix B.2, I further noticed that the "higher" and "lower" classes included wordings such as "X may increase/reduce Y". According to the Cochrane EPOC guidelines that I linked earlier, **"may" is reserved for results of low-certainty evidence**. If the SR authors happened to use the word "unclear" to describe the same low certainty, the authors themselves stated in the rebuttal for **ID 3** that the result would be "unambiguously" labelled as an uncertain effect.
> > > >
> > > > The models are never told these arbitrary design choices. They don't know what the benchmark authors mean by "higher", "lower", "no difference", "uncertain effect", or "insufficient data". It is not obvious how a model can know what exactly "higher" means in this benchmark. How does it differ from "uncertain effect"? Or maybe "no difference"?

---

> > > > > ### Comment · Reviewer_bvug · 2025-08-06
> > > > > **Reviewer response (part 2)**
> > > > >
> > > > > (continued)
> > > > > ### Why do I think this is so important?
> > > > >
> > > > > The annotations seem to heavily rely on simple keyword matches (e.g., "comparable to", "unclear") that appear extremely sensitive to linguistic choices by the SR authors. They introduce a potentially high degree of randomness in the ground truth.
> > > > >
> > > > > As a result, to get every example right, a model has to a) determine the benchmark creators' implicit non-standard criteria for categorising SR results and b) read the mind of each SR's authors to detect their linguistic preferences.
> > > > >
> > > > > It is thus difficult to judge whether the failed examples are due to actual errors by the model (or relatively expert humans, for that matter, who perform roughly equivalently to the model) or because the SR authors happened to use another word.
> > > > >
> > > > > I'd like to emphasise that I looked only at the first 4 examples, which already contained 2 contestable labels. This is of course a small sample size, but it raises the distinct possibility that a considerable share of the remaining errors reflects such ambiguity. This would mean that
> > > > >
> > > > > 1. the models might already be very reliable in synthesising findings across several source studies at least on examples that don't suffer from label ambiguity introduced by the choice of annotation, and
> > > > > 2. the benchmark might already saturated, because any further improvement in this case would require rightly guessing the unobserved linguistic preferences of SR authors.
> > > > >
> > > > > I want to emphasise that systematic review interpretation involves genuine complexity, and I appreciate the challenges the authors faced. I think the benchmark's premise is good and the area is important. However, I also believe that a more robust and objective labelling approach could be found **and** would benefit from more robust validation.

---

> ### Author Response · Authors · 2025-08-04
> **Response (part 2)**
>
> (continued from previous comment)
>
> **R2: "The human experts' maximum accuracy was 75% and thus actually lower than the 77.2% achieved by DeepSeek V3 on Mixed sources? This result to me reads quite different from the authors' conclusion"**
>
> While DeepSeek V3's performance on mixed sources specifically is 77.2%, this is not a fair comparison to the human performance on a random selection of questions, and thus should be compared against the overall average of only 62.4%. We further reiterate that the human evaluation was performed by isolated non-topic experts, with limited data access, no feedback, and under the rebuttal time constraints. By contrast, a full SR (what we seek to match) is collaboratively conducted by topic experts over years of effort with full data access. We thus stand by our statement that more work is still required before LLMs can reliably match the observations from **expert-conducted SRs.**
>
> We thank you again for your thorough review of our work, but remain firm in our belief that our paper delivers on our intended methodology, which we believe provides a meaningful and impactful contribution to the community. Based on the constructive discussion, we believe the concerns are still derived from misconceptions around the data source (which we tried to address), not our method. Please feel free to ask any further questions if your concerns have not been fully clarified.
>
> [1] Cochrane. Cochrane Database of Systematic Reviews: editorial policies. Cochrane Library. Accessed 2025-08-04.
>
> [2] Elicit. Elicit: The ai research assistant. (2025). Accessed 2025-08-04.
>
> [3] Hurt RT, Stephenson CR, Gilman EA, Aakre CA, Croghan IT, Mundi MS, Ghosh K, Edakkanambeth Varayil J. The Use of an Artificial Intelligence Platform OpenEvidence to Augment Clinical Decision-Making for Primary Care Physicians. J Prim Care Community Health. 2025 Jan-Dec;16:21501319251332215. doi: 10.1177/21501319251332215.
>
> [4] Alejandro Lozano, Scott L Fleming, Chia-Chun Chiang, and Nigam Shah. Clinfo. ai: An
> open-source retrieval-augmented large language model system for answering medical questions
> using scientific literature. In PACIFIC SYMPOSIUM ON BIOCOMPUTING 2024, pages 8–23.
> World Scientific, 2023.
>
> [5] Zakka C, Shad R, Chaurasia A, Dalal AR, Kim JL, Moor M, Fong R, Phillips C, Alexander K, Ashley E, Boyd J, Boyd K, Hirsch K, Langlotz C, Lee R, Melia J, Nelson J, Sallam K, Tullis S, Vogelsong MA, Cunningham JP, Hiesinger W. Almanac - Retrieval-Augmented Language Models for Clinical Medicine. NEJM AI. 2024 Feb;1(2):10.1056/aioa2300068. doi: 10.1056/aioa2300068.
>
> [6] Seidler, A. L., Hunter, K. E., Cheyne, S., Berlin, J. A., Ghersi, D., & Askie, L. M. (2020). Prospective meta‐analyses and Cochrane's role in embracing next‐generation methodologies. The Cochrane database of systematic reviews, 2020(10), ED000145.

---

> ### Author Response · Authors · 2025-08-07
> **Response (round 3)**
>
> We appreciate your continued clarification of your perspective and concerns, which we summarize here, as we believe they have been addressed:
>
> Your initial concern was “doubts about the quality of the curated questions” given that:
>
> **R2: "The answer [...] doesn't always match the underlying meta-analysis results”:**
>
> - We clarified that our annotations are consistent with the written author’s conclusion.
>
> In your second response, you argue that our source of incorrectness was instead that the SR authors' writing is inconsistent with Cochrane Guidelines (which you linked):
>
> **R2: “No matter what the SR authors' words [...] goes directly against Cochrane's own guidance. [...] ID 1 was written by two clinicians only, without any reported advice from systematic review methodologists or statisticians [...] the review text contains a questionable statistical interpretation."**
>
> - We clarified that the SR is peer-reviewed by the necessary experts (at least a statistician and leader in the field) and serves as a gold standard.
>
> You seem to now agree that the SR author wording is **acceptable, but suboptimal**. Thus, the omission of “crucial **context** that the annotators' simple word matching ignores” **is the source of your concern**.  We acknowledge that, by simplifying to discrete labels, nuance and precision is lost. However, the design choice to use discrete labels allows for scalable and consistent evaluation. To allow LLMs to only provide free-form responses, we would need manual expert grading (not scalable) or use LLMs-as-a-judge (which we believe are inadequately precise).
>
> Specifically, you argue that our omission of context **results in ambiguously-assigned Uncertain Effect labels, confusing the models**. Precisely due to the fairly standardized SR language you mention, we are confident that our labeling system is not sensitive or random as you are concerned.
>
> Firstly, during all evaluations, we have asked models to create a free text response justifying their answer, allowing us to inspect the LLM reasoning (as shown in Appendix O); our investigation shows that model failures are directly tied to interpretation of evidence, not confusion about label definitions.
>
> However, to further address your point, we also ran a small-scale experiment with the Uncertain Effect class removed. This task reduces to predicting the direction (not even size) of effect, as you suggested in your second response. Even with this simplified task, we observe that, while improved, model performance is still far from matching the SR authors.
>
> | Model                      |   Baseline |   No Uncertain Effect |   Delta |
> |:---------------------------|-----------:|----------------------:|--------:|
> | Llama 3.0 70B              |      0.368 |                 0.41  |   0.042 |
> | Qwen2.5-72B-Instruct       |      0.528 |                 0.61  |   0.082 |
> | GPT-4.1 mini               |      0.564 |                 0.643 |   0.079 |
> | Llama 3.3 70B (R1-Distill) |      0.58  |                 0.7   |   0.12  |
> | DeepSeek V3                |      0.624 |                 0.705 |   0.081 |
>
>
> To summarize, we believe that during the discussion period we have addressed most of your original concerns from the first and second response: models are provided with sufficient information, our labels match the authors' (summarized) written interpretation, and the SR authors produce valid statistical interpretations verified by a peer review process.
>
> You’ve raised new concerns of how our simplification can make evaluation ambiguous. In response, we have presented a new evaluation without the class that you raised concern for and observed similar results as in our original manuscript.
>
> While we recognize labeling approaches could always be refined, we believe that we are providing a novel and meaningful resource to the field; there is no other benchmark to evaluate this task, even though models are being used in a similar fashion by medical professionals. In light of our additional justifications and experiments, would you consider adjusting your score by at least one point to reflect the updated evaluation of our work? If not, could you please provide more details on what kind of labeling would meet your criteria?
>
> Thank you very much again for your engagement with our paper and your thoughtful comments. We are happy to discuss further!

---

### Official Review · Reviewer_XVhg · 2025-06-30

**Rating:** 4
**Confidence:** 2

**Summary:**

This paper describes a dataset, MedEvidence, which pairs findings from systematic reviews with the studies they are based on.
The authors extract individual findings and reformat them into a closed question-answering task to enable straightforward evaluation.
They benchmark 24 LLMs on MedEvidence and conclude that LLMs cannot reliably reproduce the observations made in expert-conducted systematic reviews.

**Additional Feedback:**

* Multi-step refinement is used to iteratively refine the answer based on a sequence of article chunks. Have you investigated the impact of the order in which articles are provided? For example, do later articles have a greater influence on the final answer?
* Since most systematic reviews have been published for a while, have you investigated the potential issue of data leakage? That is, LLMs might rely on their prior knowledge (from training data) rather than the provided sources to generate the correct answer.

**Dataset Code Accessibility:**

Yes

**Dataset Code Comments:**

Dataset is available at huggingface and code at GitHub

**Ethical Considerations:**

No, there are no or only very minor ethics concerns

**Final Justification:**

After reading author response, i updated the rating to 4. The proposed changes and additional details have addressed my concern.
* The authors say that they will 'clarify the scope of the question types included in our benchmark by revising the abstract'
* The authors provide additional experimental results and clinicians' performance.

**Limitations Weaknesses:**

My gut feeling is that the paper does not read like an AI conference paper, but might be more suitable for a digital health journal. There are several issues:

1. The question-answering setup may only be suitable for a particular type of systematic review in the clinical domain—those focused on treatment-outcome relationships. In many other cases, systematic reviews aim to curate structured tables (e.g., what genes have been studied, what functions have been identified, what evidence supports each function), which this setup does not seem to capture.
2. The paper reads more like a technical report, lacking careful exploration of experimental variants and in-depth insights. For example, while the authors evaluate two types of prompting setups, I am not convinced that the conclusions drawn about LLM performance reflect the models’ capabilities rather than just the effectiveness of specific prompt designs.
3. One major challenge in systematic reviewing is handling large volumes of literature, including very long documents from diverse sources. However, the strategy for processing long texts in this paper seems overly simplified, and no thorough investigation of alternative design choices is provided.
4. It might be worth reporting how humans with different levels of domain knowledge perform on the benchmark, as well as the effort (hours) required.

**Strengths Contributions:**

* The studied topic—exploring LLMs for systematic reviewing—is an important research area with potential practical impact.
* The authors detail the dataset curation process, which may help other researchers reproduce the work or create similar resources.

---

> ### Author Rebuttal · Authors · 2025-07-31
>
> Thank you for your thorough review, insightful comments, and constructive feedback. We are glad you recognize that our work is “an important research area with potential practical impact.”  We also appreciate your acknowledgment of the level of detail in our work, which may support reproducibility and assist other researchers in developing similar resources. We address the opportunities for improvement that you raised:
>
> **R1.1 “The question-answering setup may only be suitable for [...] treatment-outcome” SRs**
>
> We acknowledge that SRs vary in scope and structure. Given their pivotal role in guiding clinical care (especially treatment decisions informed by emerging evidence) our work focuses specifically on treatment–outcome SRs. To construct the benchmark, we were inspired by the PICO framework (Patient, Intervention, Comparison, Outcome), a widely adopted standard for generating focused, answerable clinical questions. While PICO is effective for evaluating treatment efficacy, we recognize it may not generalize well to other SR types. We explicitly state this limitation in the introduction: “Each question evaluates comparative treatment effectiveness on clinical outcomes.”
>
> **Actions:** To avoid confusion, we have clarified the scope of the question types included in our benchmark by revising the abstract. We have also expanded on this limitation in the discussion section and updated the Figure 1 description to more accurately reflect the specific question types addressed in our benchmark.
>
> **R1.2.1 “The paper reads more like a technical report, lacking careful exploration of experimental variants and in-depth insights.”**
>
> We appreciate the suggestion. We followed the DnB track guidelines and checklist, which emphasize the introduction of novel datasets. Our work focuses on creating a controlled dataset derived from studies included in existing SRs, thereby eliminating retrieval errors. We highlight that reviewers recognized the strength of our dataset curation process, describing it as “detailed” (R1) and “robust and well-documented” (R3), highlighting its potential for reproducibility.
>
> We analyze performance across 24 models, exploring the effects of model size, context length support, domain-specific (medical) finetuning, and input length (using abstract vs. full text). In total, we do provide a careful exploration through more than 10 different analyses, including:
> - Analysis by measures of evidence certainty (GRADE, source concordance) -> Fig. 5
> - Analysis of model behavior on different answer classes -> Fig. 4b
> - Analysis of question difficulty -> Appendix L
> - Analysis by number of sources -> Appendix F
> - Analysis by number of tokens -> Appendix G
> - Qualitative exploration of model behavior -> Appendix O
> - Performance as a function of medical specialty -> Appendix M
>
> **R1.2.2 “I am not convinced that the conclusions drawn about LLM performance reflect the models’ capabilities rather than just the effectiveness of specific prompt designs.”**
>
> We do acknowledge that multiple reviewers have concerns regarding evaluation regimes, which we have addressed by adding:
> 1. Few-shot (1-shot, 2-shot, and 3-shot) prompting analysis.
> 2. Evaluation without chain of thought.
>
> Thus, in total, we evaluate four different techniques (zero shot, few shot + CoT, zero shot + CoT, expert-guided prompting). While a few models improve their rankings under the new few-shot evaluation, the overall findings remain consistent: Across all models, we observe high error rates; zero-shot DeepSeek V3 remains one of the best models; and a performance gap between LLMs and human experts still persists. Similarly, omitting CoT does not provide any statistically-significant difference.
>
> | Model                      |   With CoT |   No CoT |   Delta |
> |:---------------------------|-----------:|---------:|--------:|
> | Llama 3.0 70B              |      0.368 |    0.36  |   0.008 |
> | Qwen2.5-72B-Instruct       |      0.528 |    0.512 |   0.016 |
> | GPT-4.1 mini               |      0.564 |    0.564 |   0     |
> | Llama 3.3 70B (R1-Distill) |      0.58  |    0.576 |   0.004 |
> | DeepSeek V3                |      0.624 |    0.604 |   0.02  |
>
>
> | Model | 0-shot | 1-shot | 2-shot | 3-shot |
> |:--------------------------|--------:|:---------------|:---------------|:---------------|
> | Llama 4 Maverick | 0.448 | 0.583 (+0.135) | 0.583 (+0.135) | 0.632 (+0.184) |
> | Llama 3.3 70B-Instruct | 0.492 | 0.555 (+0.063) | 0.571 (+0.079) | 0.579 (+0.087) |
> | GPT-4.1 mini | 0.564 | 0.619 (+0.055) | 0.595 (+0.031) | 0.619 (+0.055) |
> | Llama 3.3 70B (R1-Distill) | 0.58 | 0.599 (+0.019) | 0.591 (+0.011) | 0.591 (+0.011) |
> | DeepSeek V3 | 0.624 | 0.567 (-0.057) | 0.555 (-0.069) | 0.575 (-0.049) |
>
> **Actions:** We have updated the discussion to add all our different evaluation techniques. We have also expanded the limitations section to include potential methods for improving our task, such as fine-tuning.
>
> **R1.3 ”One major challenge in systematic reviewing is handling large volumes of literature [...]  the strategy for processing long texts in this paper seems overly simplified, and no thorough investigation of alternative design choices is provided.”**
>
> We appreciate the concern. To address long-context limitations, we explored two strategies: using abstracts only and a naïve summarization approach. While the latter helps short-context models, it can hurt long-context models (Appendix Fig. 20). This analysis was motivated by our inclusion of smaller models; however, 80% of questions require ≤15K tokens—within range for 22 of the 24 models we evaluated (Appendix Table 3). Even models with 1M-token capacity (e.g., GPT-4.1) still exhibit high error rates.
>
> Scientific papers average ~6K tokens, thus we report performance across source counts (Appendix F), token counts (Appendix G), and abstract-only setups (Appendix N). A core aim of our work is to evaluate LLMs' zero-shot (and few-shot) performance given their growing clinical use. While we include one expert-guided method, fully addressing long-context reasoning warrants a dedicated investigation.
>
> **Actions:** We have updated the Results to highlight that 22 of 24 models can process 80% of the dataset and noted in Future Work that long-context modeling remains an open challenge.
>
> **R1.4 “It might be worth reporting how humans with different levels of domain knowledge perform on the benchmark, as well as the effort (hours) required.”**
>
> We appreciate your feedback and agree this is an important question. Due to time constraints, we recruited three external clinicians (with no connection to this work) with diverse specialties (e.g., pediatrics, oncology) and an average of four years of experience. Each clinician answered 20 randomly selected questions using the same inputs provided to the language models (i.e. abstract-only when models would only have abstract access). The average response time was ~10 minutes per question and the mean accuracy was 71.67% (min=65%, max=75%). Notably, every question was answered correctly by at least one clinician. Interrater agreement was a moderate Fleiss' kappa of 0.43.
>
> **Actions:**
>
> - We updated the Appendix to include this experiment and describe the evaluation protocol. We note that clinicians worked individually, were not necessarily experts in each question's topic, and could not consult external resources. This setup differs substantially from systematic reviews, which are typically conducted by domain experts over extended periods with unrestricted access to information.
> - We updated Figure 1 to include clinicians' performance under time-constrained conditions.
>
> **R1.5 “Multi-step refinement is used to iteratively refine the answer based on a sequence of article chunks. Have you investigated the impact of the order in which articles are provided?”**
>
> Shuffling the question order does not result in performance changes beyond the ranges expected by our confidence intervals; thus, our results suggest that source order has no statistically-significant effect on model performance.
>
> | Model | Original Order | Shuffled | Delta |
> |:------------------------|----------------:|---------:|------:|
> | Llama 3.0 70B | 0.368 | 0.356 | 0.012 |
> | Qwen2.5-72B-Instruct | 0.528 | 0.5 | 0.028 |
> | GPT-4.1 mini | 0.564 | 0.548 | 0.016 |
> | Llama 3.3 70B (R1-Distill) | 0.58 | 0.564 | 0.016 |
> | DeepSeek V3 | 0.624 | 0.572 | 0.052 |
>
> **R.1.6 ”Most systematic reviews have been published for a while, have you investigated the potential issue of data leakage? That is, LLMs might rely on their prior knowledge (from training data) rather than the provided sources to generate the correct answer.”**
>
> To properly evaluate the potential risk of data leakage, we perform an experiment to quantify if models are able to answer questions in our benchmark (which were newly formulated for the purpose of this work) by prompting them to answer the questions without providing any articles as context. Evidently, models perform far worse when asked to answer the questions from memory; this trend is consistent across the range of baseline performances, with Llama 3.0 70B even performing worse than random.
>
> | Model  |  With Evidence |  Memory Only |  Delta |
> |:--------|----------------:|--------------:|--------:|
> | Llama 3.0 70B  |  0.368 |  0.116 | -0.252 |
> | Qwen2.5-72B-Instruct   |  0.528 | 0.388 | -0.14 |
> | GPT-4.1 mini | 0.564 | 0.428 |  -0.136 |
> | Llama 3.3 70B (R1-Distill) | 0.58  |  0.396 | -0.184 |
> | DeepSeek V3  |  0.624 |  0.324 |  -0.3 |
>
> We thank you for your thoughtful feedback. Your suggestions have greatly helped improve the manuscript, and we’re happy to address any further questions.

---

> > ### Comment · Reviewer_XVhg · 2025-08-02
> >
> > Thanks for the detailed response, which has addressed my concerns. I will update the assessment accordingly.

---

### Note · Authors · 2025-08-16

We would like to take this opportunity to truly thank the reviewers for their valuable feedback. We are inspired that all reviewers found our work **well-motivated** and our **dataset construction details thorough** enough to facilitate reproduction and engage in productive dialogue about our curation process. Multiple reviewers' support that our benchmark has potential as a **high-quality dataset that fills a gap in a timely and relevant problem** has been greatly motivating. We are also grateful for the reviewers' additional insights and recommendations, which have allowed us to further refine the manuscript. In particular, we are glad that our clarifications and further analyses on more prompting strategies, human performance under equivalent conditions to models, and further qualitative analysis were able to **address several reviewers' concerns**, ensuring that we release a useful resource to the community. We hope that our work can, as reviewers stated, help evaluate **multi‐source evidence synthesis**, test models on **processing long-context literature**, and better **approximate real systematic review** by using original source evidence.

One reviewer has expressed concerns that our standardized classification labels limit models' ability to express nuance and evaluate results in context. We hope that our additional experiments and explanations have convinced the reviewer that **our results hold** when accounting for the lost context and that our labeling system was chosen intentionally to **enable scalable evaluation** while still using the **highest-available source of ground truth**.

We would finally like to thank the ACs for their oversight and facilitation of the review process. Again, we deeply thank everyone for engaging with us throughout the review period.

---

### Decision · Program_Chairs · 2025-09-18

**Decision:**

Reject

**Comment:**

This paper presents a dataset of findings from systematic reviews (SRs) on Cochrane paired with studies they are based on.  Each review setting is paired with questions with closed-form answers representing the human conclusions. 24 models are evaluated based on how well they can answer these questions.

Strengths
1. Topic is very important
2. Dataset is reasonably broad
3. Lots of LLMs studied
4. Interesting findings, such as more evidence sources matching the ground truth does indeed improve answering performance (although this could also
indicate easier/simpler questions)

Weaknesses
1. Restriction to a certain type of QA (and to treatment-outcome scenarios)
2. Missed opportunity to handle longer contexts
3. Potential issues in the benchmark

Discussion:

1. Issues in the benchmark are clarified. I tend to agree with the authors that Cochrane should be trusted. I think what bvug raises are interesting and real issues but that re-evaluating Cochrane is probably not the right move for dataset curation, and in general it would probably not be higher-quality to do this re-evaluation anyway.

2. Use of full text is clarified. While it would be great to have more full-text articles, and backing off to abstracts is not great, I think this is fine given the pragmatic constraints.

3. Lack of more prompts or prompting strategies. This is a fair criticism although can be applied to most papers of this form.

Conclusion

As a result of this discussion, I ultimately discounted the assessment of bvug. bvug did a good job of reading through instances from the dataset, but the authors' response has me convinced that they have considered these issues and addressed them. I see a bit of anchoring bias where bvug is sticking with the initial assessment despite some holes poked in their concerns. What I see here is not a deep methodological flaw but an area for improvement -- and I think accepting papers only with zero room for improvement leads to an overly conservative program. The SAC and I discussed this paper and converged on this.

I think the question for this paper is whether this dataset will be useful to others, not the depth of experiments in this paper as raised by reviewer XVhg. I think ultimately the answer is yes. This is a significant enough problem that probably it will be studied.

===== FINAL UPDATE FROM DB Track PCs ====

The final decision for this paper has been taken by the program chairs after consultation with the SACs. All Senior Area Chairs have ranked papers according to the feedback from the AC during the review process. We decided to leave the original meta-review to reflect the opinion of the AC in light of the initial discussions with reviewers and SAC.